# CAUSAL REPRESENTATION LEARNING ON DEGRADED MULTI-SENSOR STREAMS

## ABSTRACT

Many systems require real-time fusion of *multi-sensor* streams to produce causal estimates that drive online decisions. These systems must distill information across sensors while contending with missing and degraded measurements. As the number of sensors grows, both observable dropouts and latent degradation become more likely, making multi-sensor, multi-task processing brittle for conventional sequential models. We propose two *plug-in* modules that attach to any unidirectional backbone (e.g., LSTM or causal Transformer): **(i) Subchannel Hierarchical Input Embedding (SHIE)** forms channel-level embeddings from fine-grained subchannels so that degraded values perturb only a local slice of the representation; **(ii) Repetitive Cross-Modal Fusion Transformer (RCFT)** performs iterative *sensor-wise* (cross-modal) attention at each time step, fusing concurrent measurements across sensors. Both modules support many-to-many estimation and are domain-agnostic with respect to loss functions and input/output shapes. We augment vanilla LSTM and Transformer backbones with SHIE and RCFT and evaluate on four multi-sensor datasets: electric grid state estimation, physical activity monitoring, room occupancy prediction, and cognitive load estimation. Across datasets, the augmented models outperform their baselines and remain accurate as missing-data rates rise far beyond those seen in training. Ablations isolate the contribution of each module, and the combined approach improves robustness *without* relying on separate imputation.

## 1 INTRODUCTION

Modern sensing systems often operate on multi-sensor data streams Hall and Llinas (1997). Each sensor stream captures a complementary slice of the underlying dynamics; fused together, they form aligned *multi-sensor time series*. Joint analysis of these data streams can reveal structures invisible to any single sensor, enhancing performance in real-time state estimation, forecasting, control, and anomaly detection in power systems Kardakos et al. (2013); Ghasemkhani et al. (2018), traffic management Hameed et al. (2023); Kadiyala and Kumar (2014), environmental monitoring Gruca et al. (2022), and healthcare analytics Hayat et al. (2022); Li et al. (2024). Many such applications impose a strict *causality* constraint: predictions at time $t$ must be produced without access to $t' > t$, precluding retrospective smoothing.

Deploying such causal systems in practice raises two intertwined challenges—heterogeneity and degradation. **Heterogeneity:** sensor streams differ in sampling, scale, units, and physical location, spanning different sensor modalities, yet must be fused extemporaneously Nemec et al. (2016); Karle et al. (2023). **Degradation:** real sensor streams suffer from noise, corruption, and missing values. Readings vanish because of wireless outages, batteries fail, sensors drift, and low-cost loggers inject bursts of outliers. Missing-data rates routinely exceed 10–40 % in traffic monitoring Bae et al. (2018) and are comparably high in electronic-health and wearable-sensor deployments Beaulieu-Jones et al. (2018); Braem et al. (2024). Discarding corrupted segments can waste scarce information, while manual cleaning is infeasible at scale. Models must therefore remain robust when modalities or sub-channels intermittently disappear or become corrupted.

Traditional solutions perform *imputation* before learning. However, most of these methods, such as Yoon et al. (2018); Stekhoven and Bühlmann (2012); Sun et al. (2024); Cao et al. (2018); Tashiro et al. (2021); Rubanova et al. (2019), depend on information from both preceding and succeeding contexts,

making them unsuitable for application in real-time or streaming environments. Single-step fusion compounds the issue. Merging all modalities once at the input leaves subsequent layers blind to modality-specific noise, while purely recurrent backbones struggle to carry fine-grained cross-modal cues forward in time. Therefore, recent surveys call for *adaptive fusion* architectures that dynamically weight each stream throughout the network Zhao et al. (2024).

This motivates our core questions: *Can we build a fully end-to-end causal fusion model that iteratively exchanges information through layers across multiple modalities? Can this model precisely predict even with degraded sensor inputs? Can this model support streaming inference via incremental state updates?* We evaluate these questions in the context of other state-of-the-art model performances.

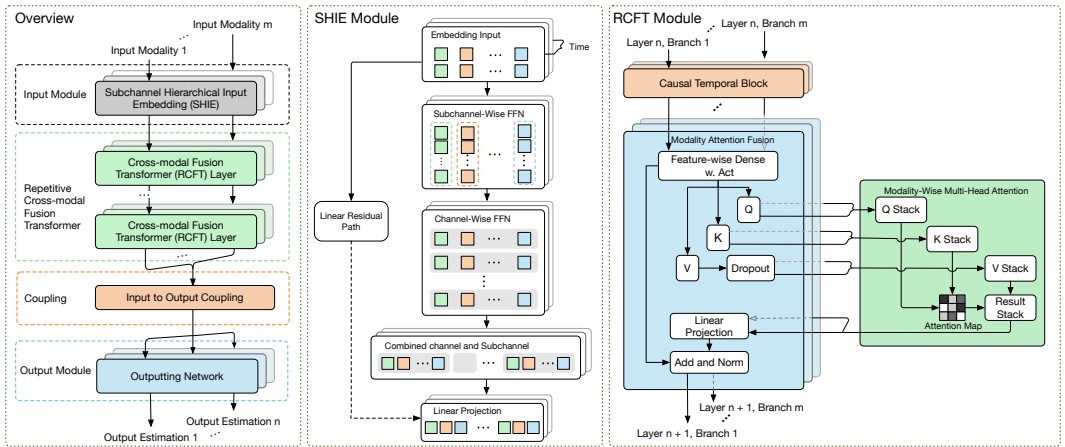

Figure 1: Architecture of Proposed SHIE and RCFT Modules

In this work, we introduce two lightweight, plug-in modules that turn a generic causal backbone into a robust multimodal estimator. **Subchannel Hierarchical Input Embedding (SHIE)** first projects every sub-sensor (for example, each axis of an accelerometer or the real and imaginary parts of a complex number) into its own latent space and then aggregates within each channel so that a missing value perturbs only a small slice of the embedding instead of the entire feature vector. **Repetitive Cross-Modal Fusion Transformer (RCFT)** maintains one branch per modality and alternates a single causal, temporal block with a modality-wise attention step at every layer, enabling each stream to iteratively refine its representation using its own history and selectively borrow cues from other streams. SHIE and RCFT can be attached independently or jointly to any unidirectional LSTM or causal Transformer. The result is a fully end-to-end model that learns reliable representations for each modality at each time step, propagates corrective signals across streams over time, and respects real-time causal constraints.

**Contributions. (i)** We introduce a causal multimodal framework that couples locality-preserving subchannel embeddings (SHIE) with layer-by-layer cross-modal fusion (RCFT). **(ii)** We benchmark robustness on four public datasets from power-grid monitoring, activity recognition, biometric cognition estimation, and building occupancy, sweeping missing-data rates up to 95%. **(iii)** The combined SHIE–RCFT module consistently outperforms vanilla LSTM/Transformer backbones, strong forecasting baselines such as iTransformer Liu et al. (2024b) and TSMixer Chen et al. (2023), and state-of-the-art non-causal models such as CroSSL Deldari et al. (2024) and MBT Nagrani et al. (2021). **(iv)** A scalability study shows that RCFT gains persist with depth, while SHIE saturates at moderate widths, allowing practitioners to trade parameters for accuracy.

## 2 PROBLEM FORMULATION AND RELATED WORK

### 2.1 PROBLEM FORMULATION

**Time-aligned Multi-sensor Inputs.** Let $M$ denote the number of input sensor modalities and $N$ the number of prediction tasks. For each modality $m \in \{1, \ldots, M\}$ we observe a vector–valued time

series, $X^{(m)}$, and for each task $n \in \{1, \dots, N\}$ we observe a the target sequence, $Y^{(n)}$:

$$X^{(m)} = \big(x_0^{(m)}, \dots, x_T^{(m)}\big), \qquad x_t^{(m)} \in \mathbb{R}^{d_n}. \qquad Y^{(n)} = \big(y_0^{(n)}, \dots, y_T^{(n)}\big), \qquad y_t^{(n)} \in \mathbb{R}^{p_m}. \quad (1)$$

All series, $x_t^{(m)}$ and $y_t^{(n)}$, are *time-aligned*. That is, each sensor modality could be collected at a different sampling rate, but must be aligned (or patched) such that $x_t^{(m)}$ is in the same instant $t$. Here $d_n$ denotes the dimensionality of the observation vector of $m$-th sensor modality.

**Missing–value Masks.** Each input vector carries an element-wise binary mask

$$M^{(m)} = \big(m_0^{(m)}, \dots, m_T^{(m)}\big), \qquad m_t^{(m)} \in \{0, 1\}^{d_n},$$

where $m_t^{(m)} = (m_{t,1}^{(m)}, \dots, m_{t,d_n}^{(m)})$. The entry $m_{t,i}^{(m)}$ indicates whether $x_{t,i}^{(m)}$ is observed or missing.

**Dynamic-window Causal Forecasting.** At each instant of evaluation $v \geq 0$, the model conditions on a trailing history of length $\tau + 1$, where $\tau \in [0, v]$ is an arbitrary parameter. We gather

$$(\mathbf{X}_{v-\tau:v}, \mathbf{M}_{v-\tau:v}) = \big\{ \big(x_{v-\tau:v}^{(m)}, m_{v-\tau:v}^{(m)}\big) \big\}_{m=1}^{M},$$

with

$$x_{v-\tau:v}^{(m)} \in \mathbb{R}^{d_n \times (\tau+1)}, \qquad m_{v-\tau:v}^{(m)} \in \{0, 1\}^{d_n \times (\tau+1)}.$$

Here $x_{v-\tau:v}^{(m)}$ is the recent window of sensor stream $m$, and $m_{v-\tau:v}^{(m)}$ is its corresponding mask. Thus, each window contains both raw values and binary masks for all $M$ sensors stream over the time span $t = v - \tau, \dots, v$. We seek a *causal* mapping

$$f_\theta : \big(\mathbf{X}_{v-\tau:v}, \mathbf{M}_{v-\tau:v}\big) \longmapsto \big(\hat{y}_v^{(1)}, \dots, \hat{y}_v^{(N)}\big), \qquad v = 0, \dots, T,$$

that uses only past and present information; no inputs or targets at times $t > v$ participate in the calculation of $\hat{y}_v$.

**Many-to-many Character.** At each time step $v$, the model outputs an $N$-tuple of vectors—one prediction per task—in a single forward pass, rather than a single scalar. This formulation supports simultaneous forecasting across multiple outputs and allows a dynamic receptive window $\tau$ without restarting from $t = 0$. In streaming deployment, only the latest prediction $\hat{y}_v$ and the model's internal state are retained, allowing constant-latency updates without reprocessing the entire history.

**Handling data degradations.** Two reliability issues are considered: *explicitly missing* values flagged by the binary mask $m_t^{(m)}$ and *unknown corruptions* that leave noisy but present readings. Both are encoded in $(\mathbf{X}_{v-\tau:v}, \mathbf{M}_{v-\tau:v})$; more details are given in Section 3.4.

## 2.2 RELATED WORK

Existing work falls into three strands—sequence forecasting, imputation, and multi-modal fusion—yet no work fully meets the demands of (or is evaluated for) *causal, streaming* inference with missing inputs. Our methods are designed to fill this gap in current related works.

**Horizon-forecasting (Extrapolation).** State-of-the-art iTransformer Liu et al. (2024b) performs fixed window-to-window forecasting via cross-dimensional attention, surpassing Autoformer Wu et al. (2021) and Informer Zhou et al. (2021). TSMixer Chen et al. (2023) follows a similar windowed formulation. UnitTime Liu et al. (2024a) mixed time series data from different domain with a language model. However, in a streaming setting—where only the *latest* value is required—the output window effectively collapses to a single step; the whole network must then be re-executed as each new observation arrives, forfeiting efficiency and temporal context. We note that RCFT and iTransformer decouple channel-wise and temporal processing, but RCFT also employs temporal attention rather than using feed-forward networks (FFN). In this way, RCFT is a natural progression of concepts introduced by iTransformer.

**Imputation Pipelines.** Two-stage schemes first complete the data, e.g., spatio-temporal cokriging Bae et al. (2018), GAIN Yoon et al. (2018), MissForest Stekhoven and Bühlmann (2012) or modality-completion networks Sun et al. (2024)—and then train a task network on the "filled" sequences. Recent work also proposed imputation methods that can be trained and inferred end-to-end

Cao et al. (2018); Tashiro et al. (2021); Rubanova et al. (2019). However, because of relying on future context, most are *non-causal* and unsuitable for real-time use. Furthermore, most imputation techniques depend on identifying the locations of missing data and, therefore, are unable to address unknown data corruptions.

**Multi-modal Fusion.** Multimodal fusion architectures such as the Multimodal Bottleneck Transformer (MBT) Nagrani et al. (2021) route each modality through a small set of *bottleneck* tokens that mediate cross-modal exchange, while perceiver-style models Jaegle et al. (2022); Hawthorne et al. (2022) first project inputs onto a latent array via cross-attention and then refine that latent with self-attention. PowerPM Tu et al. (2024) instead performs multi-sensor fusion in a task-specific domain. However, in all of these architectures, when a new observation arrives, the fused representation must effectively be recomputed, which makes them inefficient in streaming settings. Moreover, none of them are designed (or evaluated) to explicitly handle missing or corrupted inputs.

## 3 METHODOLOGY

To address the limitations of existing methods, we propose two architecture-agnostic plug-in modules: SHIE and RCFT. Both are designed for causal time-series models and can be integrated into diverse backbones. In this work, we integrate them into two backbone configurations: a unidirectional LSTM network Hochreiter and Schmidhuber (1997) and a RoPE-based causal Transformer (RoFormer) Su et al. (2024) augmented with Group-Query Attention (GQA) Ainslie et al. (2023), demonstrating their versatility across distinct causal architectures.

For exposition, we assume that all sensor modalities share a common sampling grid; when rates differ, we either (i) patch higher frequency streams down to the grid or (ii) align lower frequency streams by inserting masked slots (upsampling with missing indicators) so that all inputs end up on the same grid. Fig. 1 summarizes the full architecture: each modality passes through SHIE to produce a hierarchy-aware embedding, which then feeds stacked RCFT layers that exchange information *at each time step*. Then, the latent vectors from all modality branches are concatenated and passed to a task-specific head to produce the per-step output, while strict causal masking ensures real-time operation. Note that for non-causal applications, such as sequence classification, task-specific heads can employ global-attention pooling and switch to a non-causal backbone.

### 3.1 SUBCHANNEL HIERARCHICAL INPUT EMBEDDING (SHIE)

We represent each modality's input as a tensor $X \in \mathbb{R}^{T \times C_f \times C_s}$, where $C_f$ indexes feature channels (e.g., readings from sensors of the same type across the system) and $C_s$ enumerates subchannels within each channel (e.g., accelerometer axes, one-hot encoding of categorical values), yielding a structured per-time-step representation.

**Baseline Embedding.** Conventional methods flatten $(C_f, C_s)$ into a single vector and apply one linear projection, collapsing structure into a single latent representation. By mixing all subcomponents at once, such embeddings become highly sensitive to localized corruptions or drop-outs and discard the inherent grouping structure of the data.

**Design Motivation.** SHIE mimics the logic of a high-dimensional kernel projection: first map local structure into a higher-dimensional *latent* space, then aggregate. The difference is that the mapping is learned end-to-end and applied *independently to every subchannel*, preserving locality so that missing values perturb only their own slice of the latent representation.

**Three-stage Mapping.** SHIE follows a three-stage hierarchical mapping: (1) *Subchannel expansion*: an FFN projects each subchannel into its own high-dimensional latent space; (2) *Channel aggregation*: a second FFN compresses and combines these projections into a cohesive channel representation; (3) *Final projection*: a linear layer flattens and merges all channel embeddings into the vector per time step $E \in \mathbb{R}^{T \times C_{dm}}$, where $C_{dm}$ is the final embedding size.

Because subchannels are expanded independently before any mixing, a missing or corrupted subchannel perturbs only its corresponding slice of the learned representation $E$, substantially improving robustness to drop-outs, misreads, and burst/impulse noise.

SHIE is instantiated per modality with its own parameters and supports subchannel-level masks (binary indicators of missingness). It outputs a per-timestep, per-modality embedding, i.e., a structured representation, ready for downstream causal fusion (e.g., RCFT).

The core transformation for each time step is:

$$X_t \in \mathbb{R}^{C_f \times C_s} \xrightarrow{\text{FFN}_{\text{sub}}} Y_t \in \mathbb{R}^{C_f \times C_{ds}} \xrightarrow{\text{FFN}_{\text{chan}}} Z_t \in \mathbb{R}^{C_{dm} \times C_{ds}} \xrightarrow{\text{Flatten + Linear}} E_t \in \mathbb{R}^{C_{dm}}$$

The formal definitions, shapes, and details of the implementation are given in the Appendix A.1.

## 3.2 REPETITIVE CROSS-MODAL FUSION TRANSFORMER (RCFT)

**Baseline Fusion.** Conventional multi-modal pipelines typically apply a single fusion step. In *early fusion*, all modality embeddings are concatenated at the input, forcing subsequent layers to operate on a coarse joint *representation* that can entangle and obscure modality-specific structure. In *late fusion*, each modality is processed in isolation and merged only at the output, precluding intermediate cross-modal interactions and limiting the formation of shared latent representations.

**Motivation.** To mitigate the shortcomings of early/late fusion, RCFT enables *iterative* cross-modal exchange of representations: at every depth, modalities communicate, then refine their own latent representations locally. This design is inspired by message-passing networks—passing information across branches while preserving modality-specific identity via residual updates.

**Layer Structure.** RCFT maintains separate branches for each modality and alternates causal temporal processing with modality-wise fusion at every layer. Each RCFT layer comprises: (1) *Causal temporal block:* a causal sequence model (e.g., a Transformer self-attention layer, an LSTM cell, or another strictly causal architecture) that updates each branch using information from past observations. (2) *Modality-wise fusion:* multi-headed attention over keys and values pooled from all branches, enabling each modality to selectively incorporate information from others.

By stacking $L$ such layers—each combining a temporal update and a cross-modal exchange—RCFT enables each modality to iteratively refine its representation while integrating complementary signals from others. This layered fusion avoids the brittleness of one-shot merging and helps suppress noise from corrupted inputs.

Since both its temporal and fusion operations are causally masked, RCFT preserves real-time operation and drops in to standard causal backbones (e.g., unidirectional LSTMs, causal Transformers, and their variants). It supports incremental updates, without reprocessing overlapping look-back windows at each step. The formal definitions of the layers appear in Appendix A.2, and Appendix A.3 provides the rationale of the design and compatibility analysis.

## 3.3 COUPLING AND OUTPUT

Because the set of input modalities rarely matches the dimensional and semantic requirements of the task outputs, we add a *coupling layer* directly before the output network. In this work we concatenate the per-modality embeddings at each time step and pass the result through a lightweight feed-forward network.

The subsequent output network mirrors SHIE in reverse: it projects the latent vector back to the target structure, reinstating channel and subchannel axis as required by the task. For sequence-to-sequence tasks each time step is decoded independently, whereas for sequence-level prediction we apply an attention-weighted pooling across time to form a single summary vector before classification or regression.

## 3.4 SIMULATING DATA DEGRADATIONS IN TRAINING AND EVALUATION

Real-world sensor streams are often degraded by drop-outs and distortions caused by environmental interference and communication faults Wang et al. (2024b); Ghasemkhani et al. (2018). To emulate these conditions—and to evaluate all models under a consistent stress scenario—we inject synthetic noise during *both* training and evaluation using a data-agnostic, *missing-completely-at-random* (MCAR) policy applied uniformly across all datasets. While a more realistic missing-at-random (MAR) scheme would require dataset-specific missingness statistics (and thus vary by application),

MCAR provides a controlled, reproducible benchmark for testing robustness to missing or corrupted data across diverse datasets.

We introduce three types of random perturbations: **Explict Missing Data**, **Hidden Data Corruption**, and **Background Noise Perturbation**. These perturbations are applied independently of feature values (i.e., under MCAR assumptions) but are temporally clustered: when a time step is affected, its neighbors are more likely to be corrupted as well. This reflects the burst-like failure patterns common in real deployments Sharma et al. (1998). Full sampling procedures and thresholds are detailed in Appendix B. While primarily intended to simulate sensor degradation, this scheme can also act as a form of **data augmentation**, improving generalization even on clean evaluation sets.

## 4 DATASETS AND EXPERIMENTS SETUP

### 4.1 DATASETS

We evaluate SHIE and RCFT—individually and in combination—across four tasks spanning physical, physiological, and environmental domains. All models are trained and evaluated under identical settings. Dataset modalities and input/output dimensions are detailed in Appendix D.

**Grid Voltage Estimation.** This task estimates the complex voltage state (real and imaginary components) at each node in an electric grid. Inputs include low-precision voltage magnitude/phase readings from selected sensors and full-system complex power measurements. We use two public datasets provided by NREL Palmintier et al. (2020), derived from real-world 84-node and 4583-node unbalanced power distribution systems. Each sequence contains 4320 time steps sampled every 15 minutes. Topological changes caused by switch reconfigurations are embedded in the data.

**Physical Activity Monitoring.** The PAMAP2 physical activity monitoring data set is a public wearable sensor benchmark. It contains multi-sensor time series recordings from subjects performing daily activities collected with three inertial measurement units on the body Reiss and Stricker (2012). To increase the challenge and realism of this dataset, we focus on wrist-worn sensors only, as wrist-worn biometric sensors are widely adopted. Thus a subset of sensors are used from PAMAP2.

**Room Occupancy Estimation.** This task predicts the number of occupants in a room based on non-intrusive environmental sensors. Inputs include temperature, illumination, sound, carbon dioxide, and passive infrared (PIR) data. We use the dataset from Singh et al. (2018), which spans three data collection sessions from Dec 2017 to Jan 2018. We resample all sensor readings to a consistent 30-second interval.

**Cognitive Load Classification.** The goal of this dataset is to classify subjectively reported cognitive workload from aviators based on wrist-worn biometric sensors. Each input sequence includes photoplethysmography (PPG), electrodermal activity (EDA), acceleration, and skin temperature, collected using the Empatica E4 device. The dataset includes binary labels (high vs. low load), derived from NASA-TLX surveys reported post-task. We used 446 flight sessions from 40 pilots in Wilson et al. (2021), with a 5-fold stratified cross-validation to balance the pilot and label distribution. We note that another pupil-based sensor was collected via eye tracking in Wilson et al. (2021), but is not publicly available, and thus not included in this study.

### 4.2 GENERAL TRAINING SETUP

We compare standard LSTM and Transformer baselines against our proposed modules: SHIE only, RCFT only, and SHIE–RCFT (which combines both). All Transformer variants employ causal masking to preserve real-time inference constraints, and the LSTM baseline is configured as a unidirectional chain to match this requirement. For fairness, all models are trained with identical epoch counts and learning rate schedules, and share similar depth and width across variants.

To handle known missing values, each input subchannel includes a binary mask bit indicating whether the data in that channel is observed or missing. For all datasets, $5\%$ of the inputs are randomly masked using an MCAR policy during training. Additional noise types are injected independently, as detailed in Appendix B.

## 5 RESULTS AND ANALYSIS

### 5.1 METRIC AND VALIDATION SETUP

To assist in model comparison, we introduce a succinct measure of robustness under input loss. In vision, robustness is often summarized by mCE Hendrycks and Dietterich (2019) on ImageNet-C, which averages error over a fixed catalog of corruption types with preset severities. Our focus is to quantify robustness across levels of MCAR missingness. To this end, we introduce the *Missingness Robustness Area Under Curve (MR-AUC)*, which summarizes model performance over a sweep of missing ratios.

**MR-AUC Definition.** Let $S(m)$ denote the metric value at missing ratio $m \in [0, 1]$ under MCAR, and let $\alpha \in (0, 1]$. We define the MR-AUC up to $\alpha$ as the length-normalized area under the missingness–performance curve:

$$\text{MR-AUC}(\alpha) := \frac{1}{\alpha} \int_0^\alpha S(m) \, dm. \tag{2}$$

MR-AUC shares the same semantics as the base metric and equals the expected score when the missing ratio is drawn uniformly from $[0, \alpha]$. See Appendix C.1 for the discrete implementation of this metric.

**Validation Setup.** We sweep the missing data ratio under the MCAR policy from the $5\%$ training setting up to $95\%$ in increments of $5\%$, while keeping all other noise types identical to training (see Appendix B). For all datasets, results are reported as the median over 30 runs to capture the performance distribution. Run-to-run variability is generally small, except for the cognitive load dataset, where variability is more pronounced, indicating a higher sensitivity to training data selection. For this data set, an evaluation is performed using 5-fold cross-validation on top of repeated runs 30.

For comparison, we report $\text{MR-AUC}(\alpha)$ at $\alpha \in \{0.25, 0.50, 0.75\}$ to characterize *mild*, *mid*, and *severe* MCAR degradation. We present results for SHIE–RCFT models on LSTM and Transformer backbones in Table 1, with full results and figures provided in Appendix F.3.

Table 1: MR-AUC across datasets. Rows indicate the MR-AUC level $\alpha \in \{0.25, 0.50, 0.75\}$. We compare SHIE–RCFT (S–R) on LSTM (L) and Transformer (T) backbones against SOTA baselines. Metrics marked with * are scaled by $\times 10^2$. Full results are provided in Appendix F.3.

| $\alpha$ | | **Grid Voltage Estimation (84-node)** | | | | **Physical Activity Monitoring** | | | |
|---|---|---|---|---|---|---|---|---|---|
| | Metric ↓ | S-R (L) | S-R (T) | TSMixer | iTrans. | Metric ↑ | S-R (L) | S-R (T) | TSMixer | iTrans. |
| 0.25 | | 0.404 | **0.296** | 3.380 | 2.369 | | 0.854 | **0.864** | 0.817 | 0.824 |
| 0.50 | $A$-MAE* | 0.909 | **0.711** | 5.956 | 4.011 | Accuracy | 0.808 | **0.815** | 0.753 | 0.749 |
| 0.75 | | 1.719 | **1.191** | 9.861 | 5.991 | | 0.699 | **0.680** | 0.667 | 0.646 |
| 0.25 | | 0.308 | **0.265** | 0.985 | 1.623 | | 0.659 | **0.660** | 0.654 | 0.655 |
| 0.50 | $\theta$-MAE* | 0.446 | **0.375** | 1.996 | 2.750 | AU-ROC | 0.655 | **0.655** | 0.643 | 0.647 |
| 0.75 | | 1.251 | **0.581** | 3.324 | 4.352 | | **0.632** | 0.619 | 0.624 | 0.628 |

| $\alpha$ | | **Room Occupancy Estimation** | | | | **Cognitive Load Classification** | | | |
|---|---|---|---|---|---|---|---|---|---|
| | Metric | S-R (L) | S-R (T) | TSMixer | iTrans. | Metric ↑ | S-R (L) | S-R (T) | CroSSL | MBT |
| 0.25 | | 0.019 | **0.016** | 0.337 | 0.143 | | 0.645 | **0.712** | 0.584 | 0.663 |
| 0.50 | $E$-MAE ↓ | 0.027 | **0.023** | 0.868 | 0.159 | Accuracy | 0.628 | **0.680** | 0.563 | 0.614 |
| 0.75 | | 0.036 | **0.031** | 1.232 | 0.128 | | 0.604 | **0.653** | 0.552 | 0.562 |
| 0.25 | | **0.984** | 0.980 | 0.901 | 0.910 | | 0.718 | **0.806** | 0.622 | 0.697 |
| 0.50 | Accuracy ↑ | **0.980** | 0.976 | 0.801 | 0.907 | AU-ROC | 0.716 | **0.792** | 0.606 | 0.684 |
| 0.75 | | **0.977** | 0.973 | 0.707 | 0.925 | | 0.705 | **0.770** | 0.594 | 0.664 |

**Notation:** $A$—voltage magnitude; $\theta$—voltage phase angle; $E$—expected occupant count; Accuracy denotes top-1 accuracy. Arrows: ↑ = higher is better, ↓ = lower is better. See Appendix C for metric definitions.

### 5.2 COMPARISONS ON CAUSAL PREDICTION TASK

We benchmark against the state-of-the-art methods TSMixer and iTransformer, which are originally designed for long-horizon forecasting. We leave their architectures unchanged and append a single linear projection to produce a one-step estimate (see Section 2.2 and Appendix A.3).

**Grid Voltage Estimation.** From Table 1, both of our models outperform SOTA baselines on the 84-node system across all three missingness conditions. TSMixer and iTransformer degrade rapidly as the missingness ratio increases, while the Transformer backbone achieves the strongest results among our models. For both voltage magnitude and phase angle of the complex voltage, metric definitions are provided in Appendix C.

The 4583-node system is overall less challenging, with lower loss across all methods. As shown in Fig. 7, iTransformer performs well on raw inputs but collapses almost immediately once missingness exceeds 5%. Full results are provided in Appendix F.3.1.

Finally, a per-node residual analysis—capturing mean bias and variance across all samples—confirms these trends and highlights even larger gaps among models than suggested by aggregate metrics (Appendix F.2, Figure 4-5).

**Physical Activity Monitoring.** From Table 1, SHIE–RCFT–Transformer attains the highest *Accuracy* across all missingness levels and the best *AU-ROC* at $\alpha \in \{0.25, 0.50\}$. Under severe missingness ($\alpha = 0.75$), AU-ROC leadership shifts to SHIE–RCFT–LSTM, while SHIE–RCFT–Transformer remains comparable to the strongest baseline. Overall, SHIE–RCFT variants outperform SOTA baselines (TSMixer, iTransformer) on *Accuracy* and on *AU-ROC* at mild/mid missingness. See Appendix F.3.2 and Fig. 8 for detailed curves.

**Room Occupancy Estimation.** From Table 1. SHIE–RCFT–Transformer achieves the lowest $E$-MAE across all missingness levels, while SHIE–RCFT–LSTM attains the highest Accuracy. Both variants outperform prior SOTA (TSMixer, iTransformer) under every missingness setting and metric evaluated. Appendix F.3.3 and Fig. 11 provide detailed curves, showing that iTransformer and TSMixer degrade sharply as missingness increases.

## 5.3 COMPARISONS ON NON-CAUSAL PREDICTION TASK

For non-causal prediction tasks, we benchmark our method against state-of-the-art baselines MBT Nagrani et al. (2021) and CroSSL Deldari et al. (2024) for sequence classification. MBT is a *general multimodal* architecture that we adapt to a multi-sensor setting; CroSSL targets robustness to missingness via self-supervised learning, making both approaches natural comparators under missingness.

**Cognitive Load Classification.** Across this task, our models outperform the baselines. In particular, SHIE–RCFT–Transformer is consistently stronger than SHIE–RCFT–LSTM at all missing levels for both accuracy and AU-ROC. As detailed in Fig. 12, SHIE–RCFT–Transformer degrades gradually and maintains a clear margin. CroSSL and MBT show persistently low AU-ROC and a noticeable decline in Accuracy as missingness increases.

Across datasets, our method at 30–40% missingness matches or exceeds SOTA at 0% missingness. For example, Fig. 12 shows that the SOTA performance of other models under ideal imputation (0% missingness) is matched by our model even at $\approx 30\%$ missingness. Given that real-world imputation is rarely perfect, the practical advantage of our approach is likely understated in this comparison. We also adapted TSMixer for sequence classification; it produces strong results but still falls short of our method, with detailed numbers reported in Appendix F.3.

## 5.4 ABLATION AND SCALABILITY STUDY ON SHIE AND RCFT MODULES

**Ablation Study.** We evaluate the effect of enabling or disabling SHIE and RCFT on both causal backbones (unidirectional LSTM and Transformer). Missingness levels are matched in training and evaluation at 5% for all datasets except room-occupancy. For room-occupancy tasks involving imputation, the masking rate is increased to 40% to create a more challenging setting. Table 2 presents the *in-distribution* ablation. Comprehensive ablations across missingness levels, together with SOTA comparisons and figures, are provided in Appendix F.3. From these results, we conclude that **RCFT** is the main driver of robustness as missingness increases, while **SHIE–RCFT** attains the strongest overall curve, matching SHIE at the lowest loss while maintaining high robustness across mid and high loss.

Table 2: Comparison of results across metrics, datasets, and tasks. Metrics marked with * have been scaled by $\times 10^2$. "S-R" indicates the combined use of SHIE and RCFT modules.

| Category | Dataset | Metric | LSTM | | | | Transformer | | | |
|---|---|---|---|---|---|---|---|---|---|---|
| | | | Vanilla | SHIE | RCFT | S-R | Vanilla | SHIE | RCFT | S-R |
| Grid Voltage Estimation | 84-node | RMSE* | 4.565 | 4.544 | 0.513 | 0.514 | 5.183 | 0.711 | 0.463 | **0.430** |
| | | $A$-MAE* | 1.969 | 1.965 | 0.246 | 0.247 | 2.345 | 0.405 | 0.224 | **0.209** |
| | | $\theta$-MAE* | 0.422 | 0.426 | 0.243 | 0.246 | 1.037 | 0.379 | 0.221 | **0.207** |
| | 4583-node | RMSE* | 1.320 | 1.313 | 1.058 | **1.034** | 1.361 | 1.239 | 1.054 | 1.043 |
| | | $A$-MAE* | 0.594 | 0.592 | 0.422 | **0.416** | 0.645 | 0.560 | 0.421 | 0.427 |
| | | $\theta$-MAE* | 0.520 | 0.516 | 0.451 | **0.424** | 0.592 | 0.506 | 0.452 | 0.428 |
| Physical Activity | PAMAP2 | AU-ROC | **0.660** | **0.660** | 0.648 | **0.660** | 0.659 | **0.660** | 0.653 | **0.660** |
| | | Accuracy$_1$ | 0.856 | 0.863 | 0.805 | **0.865** | 0.863 | **0.865** | 0.817 | 0.860 |
| Room Occupancy | Prediction | Accuracy$_1$ | 0.983 | 0.980 | 0.986 | 0.986 | 0.971 | 0.982 | **0.987** | **0.987** |
| | | $E$-MAE | 0.018 | 0.022 | 0.015 | 0.015 | 0.021 | 0.021 | **0.014** | **0.014** |
| | Imputation | $I$-MAE | 32.728 | 28.886 | 12.904 | 4.475 | 31.173 | 20.256 | 8.773 | **3.607** |
| | Imputation & Prediction | $I$-MAE | 31.777 | 25.537 | **1.505** | 1.811 | 26.082 | 26.034 | 8.127 | 1.537 |
| | | Accuracy$_1$ | 0.865 | 0.865 | 0.873 | 0.873 | 0.865 | 0.865 | **0.879** | 0.873 |
| Cognitive Load | – | AU-ROC | 0.717 | 0.746 | 0.724 | 0.719 | 0.608 | 0.614 | 0.730 | **0.810** |
| | | Accuracy$_1$ | 0.644 | 0.681 | 0.652 | 0.642 | 0.555 | 0.573 | 0.677 | **0.718** |

**Notation:** $A$ = voltage magnitude; $\theta$ = voltage phase angle; $E$ = expected occupant count; $I$ = imputation; Accuracy$_k$ = Top-$k$ accuracy. Detailed explanation see Appendix C.

**Scalability Study.** We test model scale on Grid Estimation 84-node dataset and observe that performance continues to improve as SHIE width and RCFT depth are scaled within the observed range, confirming that both modules contribute positively under scaling (see Appendix F.1).

# 6 CONCLUSION

We introduced two plug-in modules for *causal*, multi-sensor time-series modeling: *Subchannel Hierarchical Input Embedding* (SHIE) and the *Repetitive Cross-Modal Fusion Transformer* (RCFT). Both are lightweight, domain-agnostic components that attach to standard causal backbones (unidirectional LSTM and Transformer) without violating real-time inference constraints. To quantify robustness under input loss, we proposed *Missingness Robustness AUC* (MR-AUC), and we conducted comprehensive assessments and comparisons to strong baselines across four datasets.

**Key Findings.** **(1)** RCFT is the primary source of robustness as missingness increases and continues to benefit from greater depth. **(2)** SHIE complements RCFT by improving accuracy and stabilizing behavior under low-to-moderate loss, with gains saturating at moderate widths. **(3)** The combined SHIE–RCFT consistently provides the best accuracy–robustness trade-off across backbones and datasets, with SHIE–RCFT–Transformer particularly strong overall.

**Limitations and Future Work.** (i) We assume sensor streams are alignable to a common uniform grid (rate mismatches handled via patching and masks); extending SHIE and RCFT to truly irregular or asynchronous, grid-free event streams remains open for future work. (ii) Applications to inputs such as audio, video, or long-horizon forecasting are possible but remain unexplored here. (iii) We evaluated MCAR only; MAR/MNAR remain untested here due to the lack of missing-pattern statistics in these datasets.

In summary, SHIE and RCFT convert standard causal sequence models into robust, real-time *multi-sensor* estimators. These predictors are simple, end-to-end, and can be readily characterized. They remain reliable under distribution shifts to loss levels far beyond those seen in training, without requiring separate imputation or short-horizon extrapolation stages. This enables deployment as drop-in predictors for sensor-stream estimation in *mission-critical and safety-critical* systems, where off-nominal conditions and degraded-mode operation are expected.

## REPRODUCIBILITY STATEMENT

We provide all details needed for full reproducibility. The methodology appears in Section 3, with formal layer definitions in Appendices A.1 and A.2, and design rationale in Appendix A.3. Data sources, preprocessing steps, data split *policies*, and noise injection are described in Section 4, Appendix B and Appendix D. Results, ablations, and baseline comparisons are reported in Section 5.3 and Appendix F.3. Compute details are given in Appendices G.2 and G.3.

To support full reproduction, we release an anonymized OSF bundle containing: (i) complete source code for training, evaluation, data preprocessing and noise injection; (ii) configuration files specifying all model, training, and noise hyperparameters; and (iii) the datasets used in our experiments, so the code runs out of the box: Open Science Framework (anonymous). A public GitHub repository will be linked in the camera-ready version.

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

## A  DETAILED ALGORITHM DESCRIPTIONS

### A.1  SUBCHANNEL HIERARCHICAL INPUT EMBEDDING (SHIE)

Let the input for a single modality be

$$X \in \mathbb{R}^{T \times C_f \times C_s},$$

where:

- $T$: number of time steps,

- $C_f$: number of feature channels (e.g., sensors of the same type),
- $C_s$: number of subchannels per channel (e.g., accelerometer axes).

SHIE produces a per-modality embedding $E \in \mathbb{R}^{T \times C_{dm}}$ through a structured three-stage transformation:

$$Y = \text{FFN}_{\text{sub}}(X), \quad Y \in \mathbb{R}^{T \times C_f \times C_{ds}} \tag{3a}$$

$$Z = \text{FFN}_{\text{chan}}(Y), \quad Z \in \mathbb{R}^{T \times C_{dm} \times C_{ds}} \tag{3b}$$

$$E = \text{Linear}(\text{reshape}(Z)), \quad E \in \mathbb{R}^{T \times C_{dm}} \tag{3c}$$

Here, $C_{ds}$ denotes the latent subchannel dimension and $C_{dm}$ the final modality embedding dimension. Each modality uses its own independently parameterized FFNs. Missing subchannels may be flagged via a binary mask channel appended to $X$. All FFNs use GELU activations and normalization layers (denoted as Norm) by default.

## A.2 Repetitive Cross-Modal Fusion Transformer (RCFT)

Let $M$ be the number of modalities and $L$ the number of RCFT layers. For each layer $\ell$ and modality branch $m \in \{1, \ldots, M\}$, we maintain a hidden state

$$H_m^{(\ell)} \in \mathbb{R}^{T \times d},$$

where $d$ is the modality branch width. Each RCFT layer updates the representations through the following steps:

$$\tilde{H}_m^{(\ell)} = \begin{cases} \text{GQA}_m(H_m^{(\ell)}), & \text{for Transformers,} \\ \text{LSTM}_m(H_m^{(\ell)}), & \text{for Recurrent Networks,} \\ \text{TemporalBlock}_m(H_m^{(\ell)}), & \text{other causal time-series blocks,} \end{cases} \tag{4a}$$

$$Q_m = W_m^Q \tilde{H}_m^{(\ell)}, \tag{4b}$$

$$K_m = W_m^K \tilde{H}_m^{(\ell)}, \tag{4c}$$

$$V_m = W_m^V \tilde{H}_m^{(\ell)}, \tag{4d}$$

$$K = [K_1; \ldots; K_M], \quad V = [V_1; \ldots; V_M], \tag{4e}$$

$$A_m = \text{MHA}(Q_m, K, V), \tag{4f}$$

$$H_m^{(\ell+1)} = \text{Norm}_m(\tilde{H}_m^{(\ell)} + P_m A_m). \tag{4g}$$

**Parameter notes:**

- $\text{GQA}_m$ / $\text{LSTM}_m$: branch-specific causal temporal processing.
- $W_m^Q, W_m^K, W_m^V \in \mathbb{R}^{d \times d}$: per-branch projection matrices for queries, keys, and values.
- $\text{MHA}_m$: multi-head cross-modal attention block for modality $m$, using concatenated $K$ and $V$ across modalities. $Q_m$ is not shared across modalities.
- $P_m \in \mathbb{R}^{d \times d}$: projection matrix for attention output.
- $\text{Norm}_m$: normalization layer applied after residual fusion.

Each modality retains its own temporal encoder and attention projections to preserve modality-specific semantics. Information is exchanged through shared cross-modal attention at each layer, ensuring dynamic fusion without early entanglement. The design supports both Transformer and LSTM temporal backbones and preserves causal inference guarantees.

## A.3 Operator-Level Design Rationale and Method Compatibility

**Causal Setting.** We work in a strictly causal regime: at time $t$ the estimator may depend only on $x_{\leq t}$ (no look-ahead) and should admit *incremental* updates. Standard causal backbones (e.g., unidirectional LSTM/GRU with hidden state, causal Transformers with KV cache, causal CNN/SSM blocks) satisfy this by carrying state forward in time.

**Placement of SHIE and RCFT.** Both SHIE and RCFT are *intra–time-step* operators. SHIE forms a per-modality (pre-type of measurements) embedding $E_m(t)$ from the modality's inputs and mask at the *same* time $t$. In each RCFT layer we: (i) apply the causal temporal block in each branch to obtain $\tilde{H}_m^{(\ell)}(t)$ that already summarizes $x_{\leq t}$ via the backbone's state; (ii) perform *strictly intra-step* cross-modal fusion at time $t$ (no cross-time attention); and (iii) feed the fused representation to the next temporal block so that evidence propagates to the indices $t+n$ with $n\geq 0$ in a causal manner; there is no reverse flow from future steps ($n > 0$) back to $t$.

**Operator-level Comparison.** The methods below differ by the operators used along the *time* axis and along the *feature/modality* axis:

| Model | Time-wise operator | Feature / Modality-wise operator |
|---|---|---|
| TSMixer | FFN (across time window) | FFN (across feature axis) |
| iTransformer | FFN (across time window) | Attention (across feature axis) |
| Causal Transformer | Attention (causal) | FFN (across feature axis) |
| RCFT (ours) | Attention or Recurrent (causal) | Attention (cross-modal) + FFN (per-modality) |

Time-wise FFN mixing (as in fixed window-to-window forecasters) couples all positions within the window, so streaming inference typically recomputes the window at each step and does not yield true incremental updates. In contrast, RCFT performs synchronized fusion at the current time step and uses the causal state of the backbone, so that the updates per step remain incremental (no window recomputation) while enabling dynamic causal fusion.

Moreover, many sequence-modeling architectures operate on a single concatenated feature space rather than explicit per-modality representations; even with residuals, learned projections can entangle features and weaken modality identity. By contrast, RCFT updates each branch via the residual rule equation 4g which preserves a modality-specific identity path in representation space. This supports repeated (*repetitive*) fusion across layers without erasing modality-specific structure: cross-modal evidence is injected additively while the branch representation remains anchored, all while retaining incremental updates and causal fusion.

**Compatibility with Window Forecasters.** RCFT is designed for causal, *stateful* temporal backbones. TSMixer and iTransformer use time-wise FFN mixing over a fixed window and do not expose a per-step state, so attaching RCFT would still require recomputing the window each step and would not produce incremental inference. Making them compatible would involve replacing the time-wise FFN with a causal, state-full block - effectively changing the original backbone. Consequently, we evaluate TSMixer / iTransformer unchanged and adapt only their output to a one-step estimate, while integrating RCFT into causal backbones where its assumptions hold.

## A.4 EXPERIMENT SETUP

Within each experiment group, we use the same model configuration in all comparisons to ensure fairness. For Transformer-based models, this includes the same values of $d_{\text{model}}$, $d_{\text{ff}}$, and all Group-Query Attention (GQA) parameters. For LSTM variants, $d_{\text{model}}$ is matched to the hidden state dimension $h_{\text{hidden}}$, and all models use the same number of layers—i.e., Transformer blocks and LSTM chains are equally stacked.

## A.5 COMPLEXITY ANALYSIS

We analyze here the asymptotic time complexity of SHIE and RCFT in terms of the sequence length $T$ and the number of modalities $M$, using standard Landau $\mathcal{O}(\cdot)$ notation and treating all channel and embedding widths as constants. We assume practical applications where $M \ll T$. In a given application $M$ is determined by the sensor configuration and can be regarded as a constant.

**SHIE.** SHIE is applied independently to each modality and time step, and replaces the usual per-modality input projection with a small stack of fully connected mappings. Concretely, for each input it adds a constant number of additional fully connected layers at the input and output of the models. With all channel dimensions fixed, the cost per time step per modality is $\mathcal{O}(1)$, so over a

sequence of length $T$ with modalities $M$, the total SHIE cost is $\mathcal{O}(MT)$. Thus, SHIE contributes a term that is linear in both $M$ and $T$, and in a given application (where $M$ is constant), this becomes $\mathcal{O}(T)$. That is, the term does not change the asymptotic dependence on $T$ relative to the underlying temporal backbone.

**RCFT.** RCFT augments a causal temporal backbone with $M$ modality-specific branches and intra-step cross-modal attention. Let $C_{\text{backbone}}(T)$ denote the time complexity of a single layer of the chosen causal backbone for one sequence. A single RCFT layer then has complexity

$$\mathcal{O}\big(M\, C_{\text{backbone}}(T) + M^2 T\big),$$

where the first term comes from applying the backbone independently on each of the $M$ modality branches, and the second term comes from cross-modal attention that mixes the $M$ branches at each time step.

For the standard Transformer case considered in our main experiments, the backbone uses quadratic-time attention with $C_{\text{backbone}}(T) = \mathcal{O}(T^2)$. This yields an RCFT layer complexity of $\mathcal{O}(T^2 M + M^2 T)$. In the setup where $M \ll T$, the cross-modal term $\mathcal{O}(M^2 T)$ is dominated by $\mathcal{O}(T^2 M)$ as $T$ grows, so for a fixed application with constant $M$ the overall complexity is bounded by $\mathcal{O}(T^2)$ and therefore has the same asymptotic scaling in $T$ as a standard Transformer backbone without RCFT.

More generally, if the standard quadratic-attention backbone is replaced by a more efficient causal backbone (e.g., LSTM, linear attention or Logformer-type architectures), then $C_{\text{backbone}}(T)$ changes accordingly (for example, to $\mathcal{O}(T)$ or $\mathcal{O}(T \log T)$), and the RCFT layer directly inherits this dependence. For fixed $M$, the overall complexity in $T$ therefore matches that of the chosen backbone, and both SHIE and the cross-modal fusion term act only as constant-factor overheads relative to this underlying complexity.

## B  SENSOR DEGRADATION SIMULATION DETAILS

We introduce three types of random perturbations:

**Explicit Missing Data.** Entire measurements are zeroed out and explicitly flagged using a binary mask token, simulating known dropouts. In experiments the amount of missing data is swept from 0% to 95%.

**Hidden Data Corruption.** Large-magnitude noise is injected into randomly selected channels without masking, simulating unflagged sensor failures that the model must detect and mitigate. In experiments, corruption noise is added at a constant rate of 5% (except for raw experiments).

**Background Noise Perturbation.** Low-amplitude Gaussian noise is added to all features, simulating routine sensor imprecision and calibration drift. Background noise is added uniformly with consistent hyperparameters.

### B.1  EXPLICIT MISSING DATA

Explicit missing data is achieved by random mask-out that simulates scenarios where specific measurement values are missing but are explicitly marked as unavailable or can be easily identified as erroneous using heuristics, such as detecting values with implausible magnitudes. This type of error often arises from sensor malfunctions in a particular channel of that sensor or due to a failure in data transmission. To replicate this behavior, we randomly select certain measurement values and replace them with zero. Furthermore, to improve the robustness of the model and increase the complexity of the problem, we extend the masking process to include surrounding data, reflecting real-world scenarios where adjacent values are likely to also be affected. When masking out data, all subchannels within a channel are considered as a single group of measurements, thus, they are been

assigned with the same mask. Mathematically, this can be expressed as:

$$\mathbf{Z}_{ij} \sim \mathcal{N}(0, 1),$$

$$\mathbf{Z}'_{ij} = \frac{1}{2k+1} \sum_{n=-k}^{k} \mathbf{Z}_{(i+n)j},$$

$$\mathbf{M}_{ij} = \begin{cases} 1, & \text{if } \mathbf{Z}'_{ij} > \Phi^{-1}(\alpha; 0, \frac{1}{\sqrt{2k+1}}), \\ 0, & \text{otherwise}, \end{cases}$$

$$\text{where } i \in \{1, \ldots, T\} \text{ and } j \in \{1, \ldots, C\}.$$

(5)

**Where:**

- $\Phi^{-1}(\alpha; \mu, \sigma)$: The inverse cumulative distribution function (percent point function) of a Gaussian distribution used to determine the masking threshold.
- $\alpha$: A value between 0 and 1, indicating the proportion of data to be masked (e.g., $\alpha = 0.2$ masks 20% of the data).
- $T, C$: Represent the temporal and channel dimensions of the input matrix, respectively.
- $\mathbf{Z}_{ij}$: A random value sampled from the normal distribution $\mathcal{N}(0, 1)$.
- $k$: Defines the strength of the correlation between neighboring values. The larger $k$ is, the more continuous the masked regions become.

Finally, we apply the mask to the input features $X \in \mathbb{R}^{T \times C}$, as shown in Equation 6. If a channel is masked out, we provide an additional marker dimension in the subchannel to indicate data is missing.

$$\hat{x}_{ij} = \begin{cases} x_{ij}, & \text{if } \mathbf{M}_{ij} = 1, \\ 0, & \text{otherwise}. \end{cases}$$

(6)

### B.2 HIDDEN DATA CORRUPTION

Hidden Data Corruption is introduced by injected abnormal noise that introduces high-magnitude disturbances to specific input values, simulating significant deviations in some measurement results. Unlike random mask-out, this type of noise is not flagged as erroneous and cannot be directly identified to the model, making it more challenging for the model to distinguish between valid and corrupted data. To increase these disturbances, we employ a nonlinear multiplicative noise model Yu et al. (2019), which is known to generate data that are particularly difficult to process. This setup effectively tests the robustness of our prediction model under severely degraded conditions.

First, we sample a new mask using the same method described in equation 5 and apply it to a subset of the data. The noise is defined as:

$$\delta_{ij} = \beta \, x_{ij} \epsilon_{ij} \text{ where } \epsilon_{ij} \sim \mathcal{N}(0, 1),$$

$$\hat{x}_{ij} = \begin{cases} x_{ij}, & \text{if } \hat{\mathbf{M}}_{ij} = 1 \\ x_{ij} + \delta_{ij}, & \text{otherwise} \end{cases}$$

(7)

where $\delta_{ij}$ represents the injected noise and $\beta$ is a scaling factor controlling the noise magnitude. This ensures that the corrupted values reflect realistic but challenging scenarios for the model to learn from. $\hat{\mathbf{M}}$ is sampled differently from missing ratio $\mathbf{M}$.

### B.3 BACKGROUND NOISE PERTURBATION

Lastly, we introduce additive random noise to simulate low-level inaccuracies commonly observed in real-world measurements. Unlike abnormally injected noise, this noise is applied uniformly across all input features. The process is defined as follows.

$$\mu_j \sim \mathcal{N}(0, 1), \; \delta_{ij} \sim \mathcal{N}(\gamma \mu_j, 1), \; \epsilon_{ij} \sim \mathcal{N}(0, 1)$$

$$\hat{x}_{ij} = x_{ij} + \eta_1 \delta_{ij} + \eta_2 \mathrm{F}(\epsilon_{ij}; \sigma)$$

(8)

Where the variables are defined as:

- $\mu_j$: A random variable sampled independently for each feature $j$, representing a feature-specific offset, drawn from a standard normal distribution $\mathcal{N}(0,1)$.
- $\gamma$: A scaling factor that controls the strength of the feature-specific mean offset.
- $\eta$: A scaling factor determining the overall magnitude of the added noise.
- $F(\cdot\,;\sigma)$: A low-pass filter implemented using a Gaussian kernel with $\sigma$ as the standard deviation.

This formulation ensures that the model learns to handle small but realistic perturbations in input data while capturing potential biases across features.

These noise types are introduced at random intervals during training, ensuring that the model is exposed to various scenarios.

## C    EVALUATION METRICS

### C.1    DISCRETE IMPLEMENTATION OF MR-AUC

In Section 5.1 we defined MR-AUC as an integral over the missingness–performance curve. In practice, this curve is observed only at *discrete* missingness ratios because it is obtained by sweeping a finite set of experimental conditions (in our case, 5% increments from 0% "raw" to 95%). Accordingly, we estimate MR-AUC numerically using all sampled points up to the truncation level $\alpha$ via a trapezoidal rule.

Let the missingness ratios be a strictly increasing grid $0 = m_0 < m_1 < \cdots < m_R = \alpha$, and let $S_r := S(m_r)$ denote the score at level $m_r$. A trapezoidal discretization of Eq. equation 2 is

$$\text{MR-AUC}(\alpha) = \frac{1}{\alpha} \sum_{r=0}^{R-1} \frac{S_r + S_{r+1}}{2} \left( m_{r+1} - m_r \right).$$

If the sweep is uniform with step $\Delta = \alpha/R$ (e.g., $\Delta = 0.05$),

$$\text{MR-AUC}(\alpha) = \frac{1}{R} \left( \frac{S_0 + S_R}{2} + \sum_{r=1}^{R-1} S_r \right).$$

### C.2    METRICS USED FOR ELECTRIC GRID ESTIMATION

This section provides detailed definitions of the evaluation metrics used to assess the precision of the estimations, including RMSE of the voltage, MAE of the magnitude and MAE of the phase angle, both derived from complex voltage values (i.e., voltage phasors).

#### C.2.1    ROOT MEAN SQUARE ERROR (RMSE) OF VOLTAGE

The RMSE metric is computed by averaging the normalized root mean square error across all time steps and channels. For each channel, the squared error is normalized by the squared magnitude of the corresponding ground truth voltage, making the metric scale-invariant and comparable across nodes of varying magnitudes. A small constant $\epsilon$ is added for numerical stability.

$$\bar{V}_{\text{RMSE}} = \frac{1}{T} \sum_{t=1}^{T} \sqrt{\frac{1}{C} \sum_{c=1}^{C} \frac{\|\mathbf{y}_{t,c} - \hat{\mathbf{y}}_{t,c}\|^2}{\|\mathbf{y}_{t,c}\|^2 + \epsilon}} \tag{9}$$

#### C.2.2    MEAN ABSOLUTE ERROR (MAE) OF MAGNITUDE: $A$-MAE

The MAE of the magnitude measures the mean absolute error in the magnitude of the complex voltage values. This metric evaluates the accuracy of the predicted magnitude, while disregarding the phase components.

$$\text{Magnitude MAE} = \frac{1}{T \cdot C} \sum_{t=1}^{T} \sum_{c=1}^{C} \left| \|\mathbf{y}_{t,c}\| - \|\hat{\mathbf{y}}_{t,c}\| \right|, \tag{10}$$

where $T$ is the total number of time steps, $C$ is the number of components, $\mathbf{y}_{t,c}$ is the true complex voltage at time $t$ and component $c$, and $\hat{\mathbf{y}}_{t,c}$ is the predicted complex voltage.

### C.2.3 MEAN ABSOLUTE ERROR (MAE) OF PHASE ANGLE: $\theta$-MAE

The MAE of the phase angle measures the error in the phase component of the complex voltage estimations. It is calculated as the mean absolute error of the phase angle difference, as defined in Equation 11. The calculation accounts for cases where the target voltage magnitude is zero by applying a sign operation.

$$\phi_{t,c} = \mathrm{mod}_{[-\pi,\pi]} \left( \angle \mathbf{y}_{t,c} - \angle \hat{\mathbf{y}}_{t,c} \right)$$
$$\text{Phase Angle MAE} = \frac{1}{T \cdot C} \sum_{t=1}^{T} \sum_{c=1}^{C} |\phi_{t,c} \cdot \mathrm{sign}(\|\mathbf{y}_{t,c}\|)| \tag{11}$$

### C.3 METRICS USED FOR ROOM OCCUPANCY

### C.3.1 EXPECTED VALUE FOR PREDICTION: $E$-MAE

The prediction task involves generating a probability distribution over 8 classes, where the number of people in the room is represented as a discrete variable. The expected value of the predicted probabilities is computed as:

$$y_{\text{expected}} = \sum_{i=0}^{C} y_{\text{prob},i} \cdot i, \tag{12}$$

where $y_{\text{prob},i}$ is the predicted probability for class $i$, and $C$ is the total number of classes.

### C.3.2 LOSS FUNCTION

The total loss for this task combines the categorical cross-entropy loss ($\mathcal{L}_{\text{CCE}}$) and an expected mean squared error (MSE) term ($\mathcal{L}_{\text{MSE}}$), defined as follows:

$$\mathcal{L}_{\text{CCE}} = -\sum_{i=0}^{C} (i = y_{\text{true}}) \cdot \log(y_{\text{prob},i}),$$
$$\mathcal{L}_{\text{MSE}} = (y_{\text{expected}} - y_{\text{true}})^2, \tag{13}$$
$$\mathcal{L} = \mathcal{L}_{\text{CCE}} + \lambda \cdot \mathcal{L}_{\text{MSE}},$$

Here, $y_{\text{true}}$ is the ground truth label, and $(i = y_{\text{true}})$ is a logical expression that evaluates to 1 if $i$ matches $y_{\text{true}}$, and 0 otherwise. The term $\lambda$ is a weighting factor used to balance the contributions of the cross-entropy loss ($\mathcal{L}_{\text{CCE}}$) and the mean squared error ($\mathcal{L}_{\text{MSE}}$). In this work we select $\lambda = 0.05$.

### C.3.3 EXPECTATION MAE

We can also define the Expected Mean Absolute Error (Expected MAE) as an evaluation metric:

$$\text{Expected Occupant Count MAE} = \frac{1}{N} \sum_{n=1}^{N} |y_{\text{expected},n} - y_{\text{true},n}|, \tag{14}$$

where $N$ is the total number of samples. The Expected MAE provides an intuitive measure of the model's ability to predict the number of people, complementing the loss functions used during training.

Table 3: Summary of Dataset Features and Per–Time–Step Input/Output Dimensions

| Dataset | Grid Voltage Estimation | | PAMAP2 | Room Occupancy | Cognitive Load |
|---|---|---|---|---|---|
| | 84-node | 4583-node | | | |
| **Modalities** | Power (Complex) | | Heart Rate | Temperature | Photoplethysmography |
| | $175\times2$ | $7664\times2$ | $50\times1$ | $4\times1$ | $16\times1$ |
| | Voltage$^\dagger$ | | Temperature | Illumination | Electrodermal Activity |
| | $(38+4)\times1$ | $(1704+12)\times1$ | $50\times1$ | $4\times1$ | $1\times1$ |
| | – | – | Accelerometer | Sound | Skin Temperature |
| | – | – | $50\times3$ | $4\times1$ | $1\times1$ |
| | – | – | Gyroscope | Carbon Dioxide | Wrist Acceleration |
| | – | – | $50\times3$ | $1\times1$ | $8\times3$ |
| | – | – | Magnetometer | Passive Infrared | – |
| | – | – | $50\times3$ | $2\times1$ | – |
| **Target / Label** | State Variable (Complex) | | Activity | Occupant count | Cognitive load (binary) |
| | $195\times2$ | $8515\times2$ | 18 | 1 | 1 (sequence-level) |
| **Input Dimensions** | 392 | 17044 | 11 | 15 | 42 |
| **Output Dimensions** | 390 | 17030 | 1 | 1 | 1 (sequence-level) |
| **Total Samples** | 4320 (steps) | | 32040 (steps) | 10549 (steps) | 466 (sequences) |

**Notation.** Shapes are given as $C_f \times C_s$ per time step, where $C_f$ is the number of channels and $C_s$ the number of subchannels. $^\dagger$ Voltage inputs consist of 38 low-precision magnitude channels and 4 phase-angle channels—partial observations used alongside complex power to infer the full state. Grid targets are the full high-precision complex voltage (real and imaginary components) at every node. Cognitive Load assigns one binary label per sequence. Room Occupancy predicts the number of people at each time step. PAMAP2 predicts activity type per time step (path every 500ms).

# D    TASK AND DATASET INFORMATION

The table above summarizes the per-modality shapes for each dataset. In the following, we outline the preprocessing, sampling, and splitting procedures required for exact replication.

## D.1    GRID VOLTAGE ESTIMATION

Each sequence contains 4320 time steps sampled at 15-minute intervals. The goal is to reconstruct the full complex voltage state (real and imaginary components) at all buses, normalized to per-unit (p.u.) values based on nominal design voltages.

Input features include partial and low-precision measurements: complex power at most nodes, voltage magnitude at selected locations, and occasional phase-angle readings. The exact quantities and shapes for each modality appear in Table 3.

The dataset includes multiple system configurations caused by power switch reconfigurations (e.g., opening or closing branches). All configurations are represented in both training and test sets. To ensure a clean train-validation split, we divide training and evaluation sets along the time axis without overlapping observations.

## D.2    PHYSICAL ACTIVITY MONITORING

We use the PAMAP2 dataset Reiss and Stricker (2012). Sequences are segmented by activity following the official protocol, yielding per-subject segments; segments labeled *undefined* are discarded per the dataset guidelines. Each segment is split into two contiguous parts to create a segment-level train/evaluation split with no temporal overlap. During training, windows are sampled with a rolling scheme within the training segments. To emulate activity switches, we randomly select two segments (from the same or different subjects), concatenate them, and take a random contiguous crop with a fixed look-back length. To focus on robustness under missingness and approximate non-lab conditions, only the wrist IMU is used; all other IMUs are discarded. We then inject the noise described in

Appendix B as a data-augmentation step to simulate input loss. Finally, raw samples are aggregated into non-overlapping 500 ms patches (50 original timesteps) to define the model's time steps. See Table 3 for details.

### D.3 ROOM OCCUPANCY

The dataset comprises three distinct time periods: Dec 22–24, Dec 25–26, and Jan 10–11, 2018. Sensor timestamps alternate between 30 and 31 seconds. We resample all signals to a fixed 30-second grid using a low-pass filter (Appendix E).

### D.4 COGNITIVE LOAD

This dataset includes 446 labeled sequences from 40 unique pilots, each lasting between 73 and 425 seconds. The authors Wilson et al. (2021) collected pupillometry, photoplethysmography, wrist acceleration, peripheral skin temperature, and electrodermal activity data, but only Empatica E4 wristband related data was available and included in this work. That is, eye gaze and pupillometry data was not available. Raw signals were collected using an Empatica E4 wristband with the following sampling rates: photoplethysmography (PPG) at 64 Hz, electrodermal activity (EDA) at 4 Hz, skin temperature (ST) at 4 Hz, and wrist acceleration (WA) at 32 Hz.

To unify the temporal resolution, we group all modalities into a shared 4 Hz temporal grid (i.e., patching):

**PPG:** 64 samples per second aggregated into 16 channels, each containing one subchannel.

**EDA and ST:** naturally sampled at 4 Hz, preserved as-is with 1 channel and 1 subchannel each.

**Wrist Acceleration:** 32 Hz per axis (x, y, z), grouped into 8 temporal channels per axis (24 total), represented as 8 channels with 3 subchannels (one for each axis).

This results in the input layout shown in Table 3. Each sequence is paired with a binary cognitive load label (high vs. low), which is already provided in the dataset and derived from the NASA-TLX survey administered post-flight.

To prevent label imbalance and pilot bias, we apply stratified 5-fold cross-validation while preserving pilot identity across splits. We note that the splits used by Wilson et al. Wilson et al. (2021) were not available, and the training data selected can cause significant variations in performance. As such, a direct comparison with the results from Wilson et al. (2021) are not possible.

## E DOWNSAMPLING FILTER FOR PREPOSSESSING ROOM OCCUPANCY DATA

The Room Occupancy dataset was upsampled to a 1-second interval using interpolation to create a uniform time grid. A minimum-phase low-pass filter was then applied to attenuate high-frequency components while maintaining causality. Finally, the filtered data was downsampled at 30-second intervals to match the desired sampling rate.

To ensure effective downsampling by a factor of $30 : 1$, we designed a finite impulse response (FIR) filter with minimum-phase properties. This was achieved in two steps:

### E.1 LINEAR-PHASE FIR FILTER DESIGN

A linear-phase equiripple FIR filter was designed using the **Parks-McClellan algorithm**. This algorithm minimizes the maximum error between the designed filter and the ideal frequency response, resulting in an "equiripple" filter. The filter was designed with 1001 taps, a cutoff frequency of $\frac{1}{30}$ (normalized to the Nyquist frequency) and a narrow transition band ranging from 0.033 to 0.042 to ensure sufficient rejection of high-frequency components. The stopband attenuation of the linear-phase filter achieves approximately $-60$ dB, making it highly effective for noise suppression.

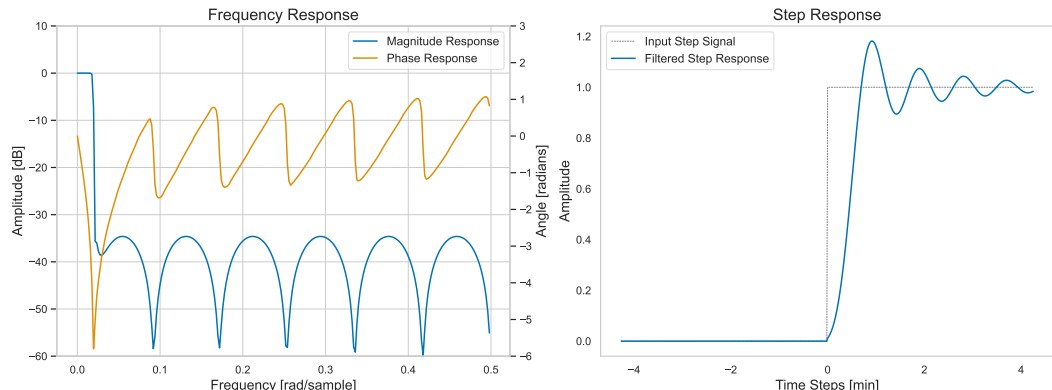

Figure 2: Frequency and Step Response of Downsampling Filter

## E.2 MINIMUM-PHASE CONVERSION

The linear-phase FIR filter was converted to a minimum-phase filter using the **Hilbert method**, a common approach for real-time systems. This conversion preserves the filter's magnitude response while minimizing group delay. The conversion also reduces the filter length from 1001 taps to 501 taps. As a result of this transformation, the stopband attenuation is reduced to approximately $-35\,\mathrm{dB}$. Given that the original data were sampled at irregular intervals of 30–31 seconds, the spectral content of high-frequency components is inherently limited. Thus, this level of attenuation is sufficient to prevent aliasing while maintaining the real-time compatibility of the downsampling process. The frequency and step response of the final filter is shown in Fig. 2.

## F ADDITIONAL ANALYSES

### F.1 MODEL SCALABILITY

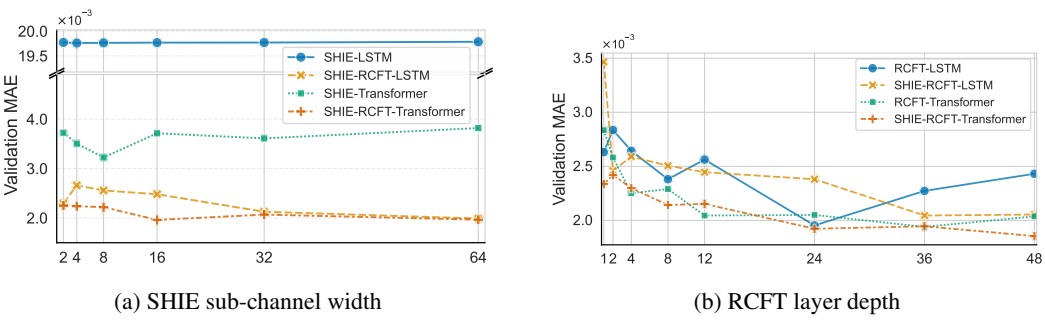

(a) SHIE sub-channel width                     (b) RCFT layer depth

Figure 3: Validation MAE on the 84-node system as SHIE width (left) or RCFT depth (right) is varied.

We use the 84-node grid estimation dataset to probe capacity because its run-to-run variance is negligible.

**Scalability of SHIE Width (Fig. 3a).** We vary the sub-channel upsampling width across $\{2, 4, 8, 16, 32, 64\}$, corresponding to expansion factors of $\{1\times, 2\times, 4\times, 8\times, 16\times, 32\times\}$ relative to the baseline subchannel width 2 respect to complex number. Fixing RCFT depth at 6 layers. In LSTM, varying the SHIE width has minimal impact alone. On Transformer, it has performance gains beyond when the unit is getting larger. For models using SHIE and RCFT jointly, performance improves slowly as SHIE width grows. This suggests that SHIE complements RCFT more effectively when tuned to moderate capacity.

**Scalability of RCFT Depth (Fig. 3b).** We fix SHIE width at 8 and scale RCFT layers from 1 to 48. On LSTM, performance improves up to 24 layers but declines thereafter, likely due to optimization difficulty or overfitting. For the Transformer, performance steadily improves through 48 layers, aided by deep normalization Wang et al. (2024a), and reaches approximately 1.7 billion parameters.

**Implications.** SHIE and RCFT shows greater performance with scaling, particularly in the Transformer backbone. These results suggest that SHIE and RCFT can be independently tuned to balance model capacity, performance, and training cost, with minimal performance degradation once the limits of the width and depth of the model are exceeded.

## F.2 RESIDUAL ANALYSIS FOR GRID VOLTAGE ESTIMATION

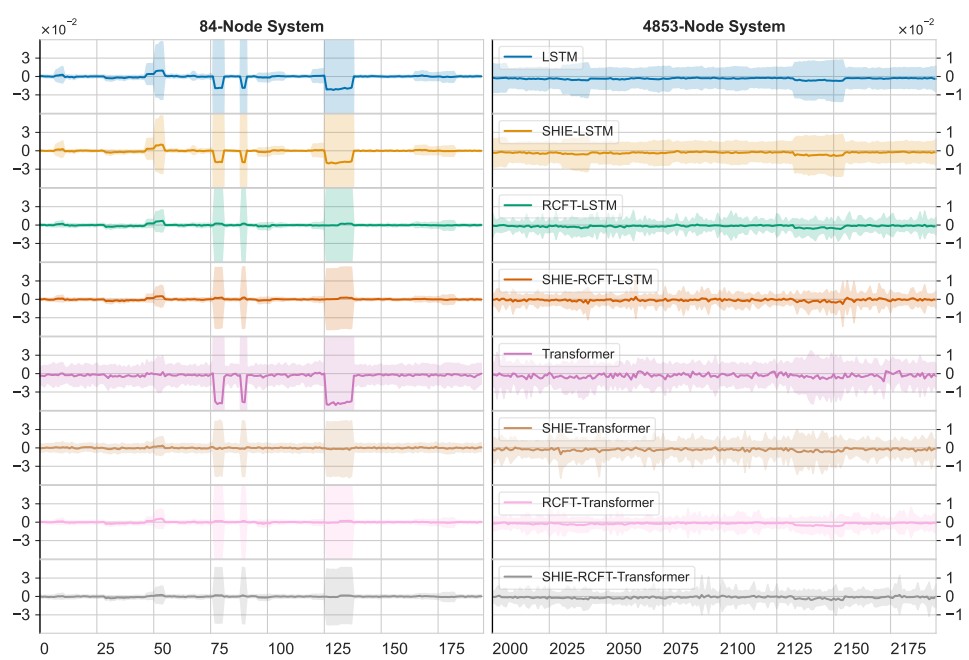

Figure 4: Per-node residuals at **5%** MCAR missingness for the 84-node (left) and 4583-node (right) systems. Each strip shows the mean residual (solid line) and $\pm 1\,\sigma$ error envelope (shaded area). The horizontal axis is bus ID; the vertical axis is residual error in voltage magnitude (per unit).

**Per-node Residuals at 5% vs. 20% MCAR.** Fig. 4 (5% MCAR, top row) and Fig. 5 (20% MCAR, bottom row) show the mean residual (solid line) and $\pm 1\,\sigma$ envelope (shaded) for each bus index. In the 84-node system, 195 bus voltages (state variables) are plotted; in the 4583-node system, a representative subset of the 8515 bus voltages is shown for clarity. The horizontal axis denotes the bus ID, and the vertical axis denotes the residual error in voltage magnitude (per unit).

**84-node.** At 5% MCAR (Fig. 4), the vanilla LSTM exhibits the widest residual envelope and three pronounced bias peaks at node indices around 50, 75–90, and 125–135. SHIE alone narrows the variance but leaves these peaks intact. RCFT suppresses the excursions and tightens the envelope, while SHIE–RCFT preserves this tight band with a slight positive mean shift.

At 20% MCAR (Fig. 5), the same bias locations persist and all models show a modestly increased variance. On the Transformer backbone, SHIE–RCFT again delivers the narrowest, flattest residuals, and SHIE–Transformer no longer exhibits a significant bias at 125–135. Within the LSTM family, the variance for vanilla and SHIE alone expands further, while RCFT alone and SHIE-RCFT display an upward mean offset rather than a downward bias. In general, SHIE-RCFT consistently provides the best trade-off between low variance and minimal bias at both noise levels.

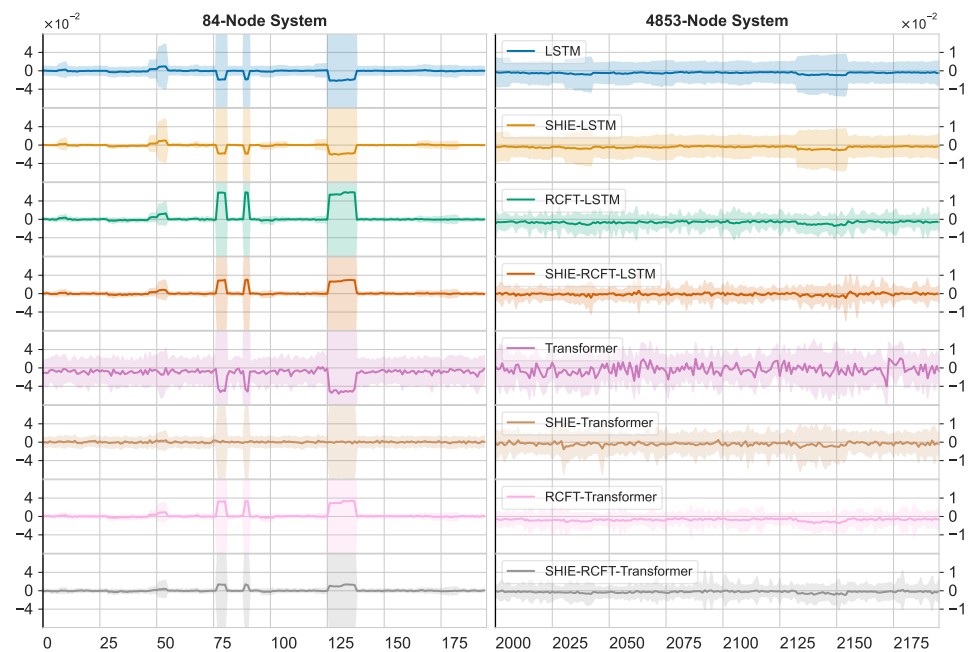

Figure 5: Per-node residuals at **20%** MCAR missingness for the 84-node (left) and 4583-node (right) systems. Each strip shows the mean residual (solid line) and $\pm 1\,\sigma$ error envelope (shaded area). The horizontal axis is bus ID; the vertical axis is residual error in voltage magnitude (per unit).

**4583-node.** Across both 5% and 20% MCAR (Fig. 4, 5), the vanilla LSTM retains the largest variance. SHIE reduces the envelope modestly, and the RCFT-only and SHIE–RCFT variants further tighten it, with SHIE–RCFT preserving a slight upward shift. Transformer backbones follow the same hierarchy: unaugmented models degrade most, SHIE reduces noise, RCFT further compresses errors, and SHIE-RCFT yields the most compact, drift-free residual distributions.

**Overall.** RCFT is the primary driver of variance reduction, while SHIE contributes additional smoothing, particularly in the Transformer backbones, without adding bias. Even when missingness quadruples from the training regime, SHIE–RCFT sustains the narrowest and most level residuals, confirming its robustness under heavier corruption.

### F.3 DETAILED MR-AUC METRICS AND MISSINGNESS-RATIO CURVES

We report robustness using MR-AUC at $\alpha \in \{0.25, 0.50, 0.75\}$, integrating the base metric of each task over missingness of MCAR up to $\alpha$ (Section 5.1). For every dataset, we run a full ablation over two modules (SHIE, RCFT) on two backbones (LSTM, Transformer), yielding {Vanilla, SHIE-only, RCFT-only, SHIE–RCFT (S–R)}, and include SOTA comparators when available (e.g., TSMixer, iTransformer, MBT, CroSSL). Task-specific base metrics are defined in the corresponding appendix sections.

Missingness is swept from $0\%$ (raw) to $95\%$ in $5\%$ increments. At each level we run **30** independent trials (different MCAR masks/seeds). For the Cognitive Load task, each trial uses **5-fold** cross-validation and we first average across the 5 folds, yielding *30 fold-averaged values* per point. The plotted curve shows the **median** of these 30 values, and the shaded band is the empirical **95% confidence interval** (2.5–97.5 percentiles). For tables, MR-AUC is computed by taking the **mean** of the same 30 values at each missingness level and then applying Eq. equation 2.

### F.3.1 GRID VOLTAGE ESTIMATION

We apply the common protocol to the 84-node and 4583-node systems. Table 4 reports MR-AUC for RMSE, $A$-MAE, and $\theta$-MAE (metric definitions in App. C.2); values are scaled by $\times 10^2$. Figure 6 plots the missingness–metric curves for the full ablation (modules on/off across LSTM

and Transformer), and Fig. 7 compares our SHIE–RCFT variants against the SOTA baselines (iTransformer, TSMixer) under the same protocol.

Table 4: Full MR-AUC results for the 84-node and 4583-node systems. All metrics are scaled by $\times 10^2$. $\alpha$ is the MR-AUC truncation parameter indicating missingness up to that level (see Sec. 5.1). "S-R" denotes the combined use of the SHIE and RCFT modules.

| $\alpha$ | Metric 84-node | LSTM | | | | Transformer | | | | TSMixer | iTrans. |
|---|---|---|---|---|---|---|---|---|---|---|---|
| | | Vanilla | SHIE | RCFT | S-R | Vanilla | SHIE | RCFT | S-R | | |
| 0.25 | RMSE | 4.565 | 4.544 | 0.697 | 0.677 | 7.483 | 1.217 | 0.616 | **0.540** | 5.375 | 4.112 |
| | $A$-MAE | 1.969 | 1.965 | 0.546 | 0.404 | 3.359 | 0.702 | 0.432 | **0.296** | 3.380 | 2.369 |
| | $\theta$-MAE | 0.422 | 0.426 | 0.324 | 0.308 | 2.125 | 0.643 | 0.290 | **0.265** | 0.985 | 1.623 |
| 0.50 | RMSE | 4.565 | 4.544 | 1.003 | 1.208 | 12.861 | 2.447 | **0.797** | 0.819 | 9.280 | 6.886 |
| | $A$-MAE | 1.969 | 1.966 | 1.010 | 0.909 | 5.835 | 1.513 | 0.869 | **0.711** | 5.956 | 4.011 |
| | $\theta$-MAE | 0.422 | 0.426 | 0.447 | 0.446 | 4.656 | 1.278 | **0.369** | 0.375 | 1.996 | 2.750 |
| 0.75 | RMSE | 4.565 | 4.544 | 2.118 | 3.559 | 21.343 | 5.192 | **1.029** | 1.784 | 15.268 | 10.266 |
| | $A$-MAE | 1.969 | 1.966 | **1.096** | 1.719 | 10.157 | 3.096 | **1.096** | 1.191 | 9.861 | 5.991 |
| | $\theta$-MAE | **0.422** | 0.426 | 0.862 | 1.251 | 9.235 | 2.771 | 0.451 | 0.581 | 3.324 | 4.352 |
| $\alpha$ | Metric 4583-node | LSTM | | | | Transformer | | | | TSMixer | iTrans. |
| | | Vanilla | SHIE | RCFT | S-R | Vanilla | SHIE | RCFT | S-R | | |
| 0.25 | RMSE | 1.320 | 1.313 | 1.089 | **1.042** | 1.573 | 1.324 | 1.088 | 1.050 | 1.584 | 1.447 |
| | $A$-MAE | 0.594 | 0.592 | 0.444 | **0.423** | 0.794 | 0.626 | 0.444 | 0.431 | 0.809 | 0.788 |
| | $\theta$-MAE | 0.520 | 0.516 | 0.451 | 0.428 | 0.738 | 0.570 | 0.450 | **0.425** | 0.667 | 0.797 |
| 0.50 | RMSE | 1.320 | 1.313 | 1.228 | 1.083 | 1.956 | 1.479 | 1.221 | **1.080** | 1.799 | 2.485 |
| | $A$-MAE | 0.594 | 0.592 | 0.549 | 0.459 | 1.042 | 0.736 | 0.547 | **0.454** | 0.958 | 1.398 |
| | $\theta$-MAE | 0.520 | 0.516 | 0.475 | 0.454 | 0.983 | 0.678 | 0.469 | **0.437** | 0.736 | 1.370 |
| 0.75 | RMSE | 1.320 | 1.313 | 1.476 | 1.200 | 2.386 | 1.655 | 1.375 | **1.172** | 2.310 | 4.269 |
| | $A$-MAE | 0.594 | 0.592 | 0.728 | 0.555 | 1.303 | 0.853 | 0.663 | **0.529** | 1.276 | 2.414 |
| | $\theta$-MAE | 0.520 | 0.516 | 0.542 | 0.521 | 1.243 | 0.794 | 0.503 | **0.493** | 0.927 | 2.326 |

**Notation:** $A$ = voltage magnitude; $\theta$ = voltage phase angle.

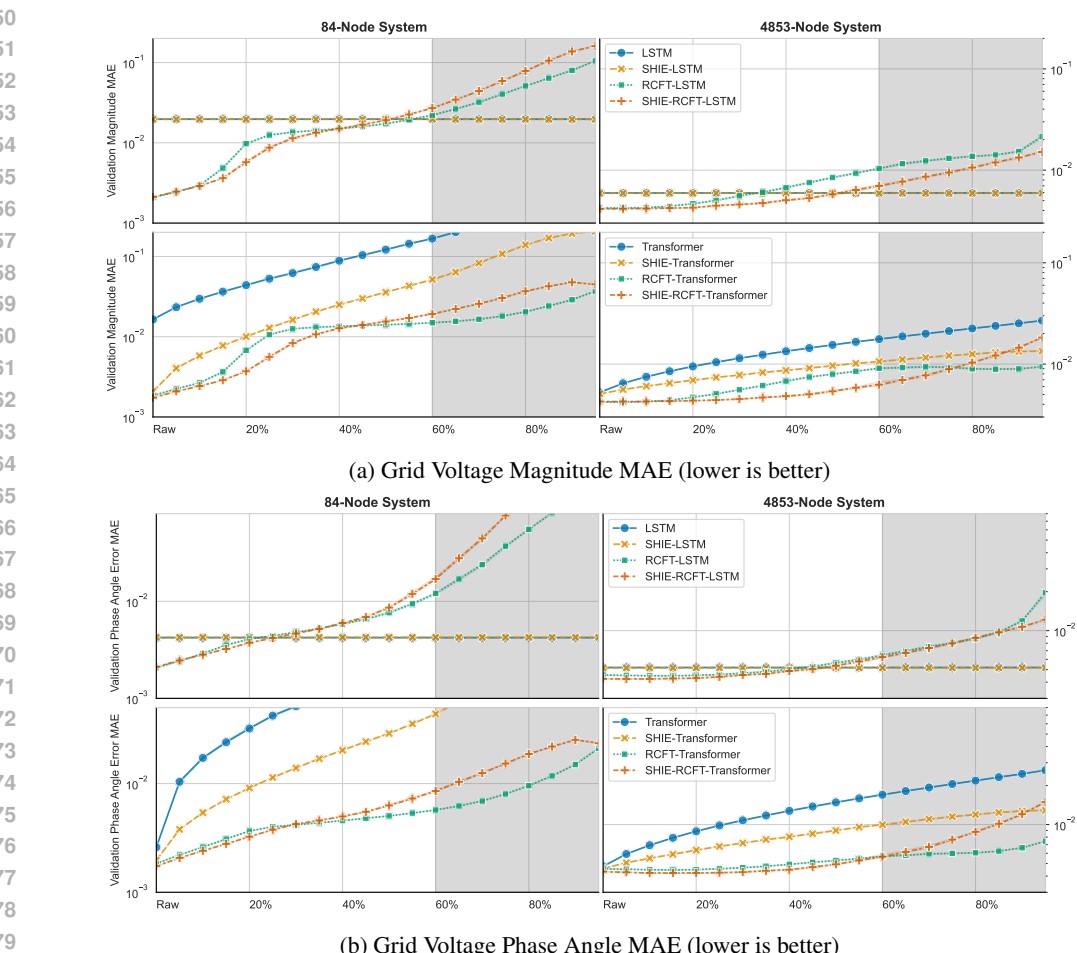

(a) Grid Voltage Magnitude MAE (lower is better)

(b) Grid Voltage Phase Angle MAE (lower is better)

Figure 6: Grid Voltage Estimation (ablation): missingness–metric curves. MCAR missingness from 0% ("raw") to 95%. "Raw" refers to original, uncorrupted sequences. In this application, magnitude is typically more important than phase angle Shirmohammadi et al. (1988); Rajicic and Bose (1988). Shaded bands denote 95% confidence intervals.

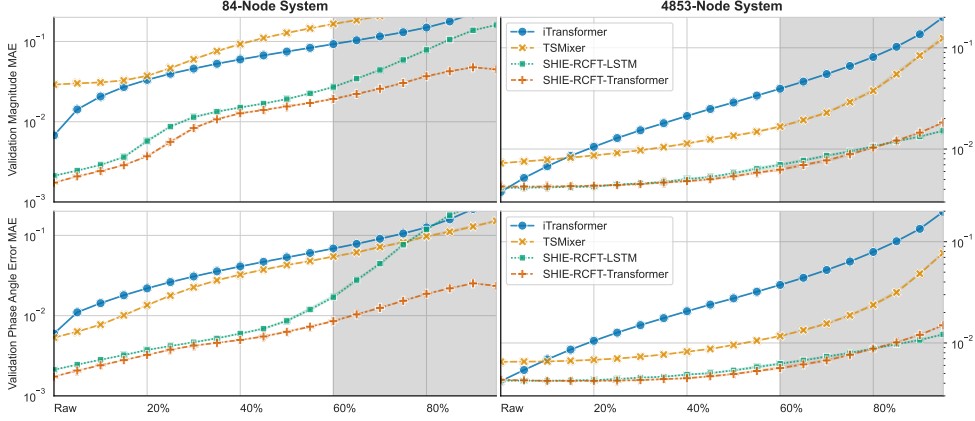

Figure 7: Grid Voltage Estimation (comparison): missingness–metric curves comparing our method to iTransformer and TSMixer (lower is better). MCAR missingness from 0% ("raw") to 95%. "Raw" refers to original, uncorrupted sequences. In this application, magnitude is typically more important than phase angle Shirmohammadi et al. (1988); Rajicic and Bose (1988). Shaded bands denote 95% confidence intervals.

### F.3.2 PHYSICAL ACTIVITY MONITORING

This appendix section provides the complete results for *Physical Activity Monitoring*. Table 5 reports MR-AUC at $\alpha$ for *Accuracy* and *AU-ROC*. Figure 9 shows the corresponding missingness–metric curves (ablation), and Fig. 8 compares our SHIE–RCFT variants to the SOTA baselines *iTransformer* and *TSMixer* under the same MCAR sweep from $0\%$ ("raw") to $95\%$.

Table 5: Full MR-AUC results for the Physical Activity Monitoring dataset. Metrics reported are Accuracy and AU-ROC. $\alpha$ is the MR-AUC truncation parameter indicating missingness up to that level (see Sec. 5.1). "S-R" denotes the combined use of the SHIE and RCFT modules.

| $\alpha$ | Metric | LSTM | | | | Transformer | | | | TSMixer | iTrans. |
|---|---|---|---|---|---|---|---|---|---|---|---|
| | | Vanilla | SHIE | RCFT | S-R | Vanilla | SHIE | RCFT | S-R | | |
| 0.25 | Accuracy | 0.819 | 0.845 | 0.795 | 0.854 | 0.842 | **0.864** | 0.806 | 0.858 | 0.817 | 0.824 |
| | AU-ROC | 0.655 | 0.658 | 0.647 | 0.659 | 0.657 | **0.660** | 0.652 | 0.659 | 0.654 | 0.655 |
| 0.50 | Accuracy | 0.689 | 0.764 | 0.754 | 0.808 | 0.767 | 0.814 | 0.765 | **0.815** | 0.753 | 0.749 |
| | AU-ROC | 0.633 | 0.646 | 0.641 | **0.655** | 0.645 | 0.654 | 0.647 | **0.655** | 0.643 | 0.647 |
| 0.75 | Accuracy | 0.566 | 0.627 | 0.657 | 0.699 | 0.657 | 0.665 | 0.673 | **0.680** | 0.667 | 0.646 |
| | AU-ROC | 0.598 | 0.616 | 0.617 | **0.632** | 0.615 | 0.619 | 0.627 | 0.628 | 0.624 | 0.628 |

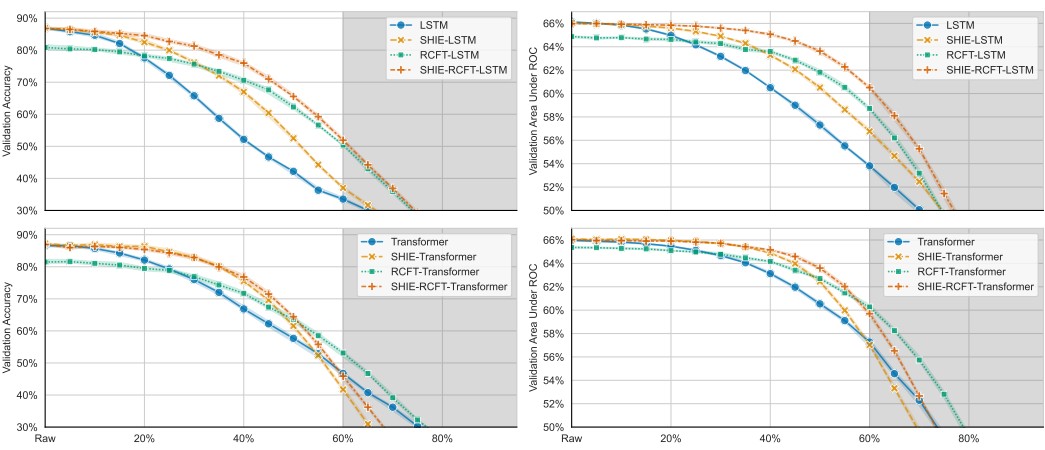

Figure 8: Physical Activity Monitoring (comparison): missingness–metric curves comparing our method to iTransformer and TSMixer (higher is better). MCAR missingness from $0\%$ ("raw") to $95\%$. "Raw" refers to original, uncorrupted sequences. Shaded bands denote $95\%$ confidence intervals.

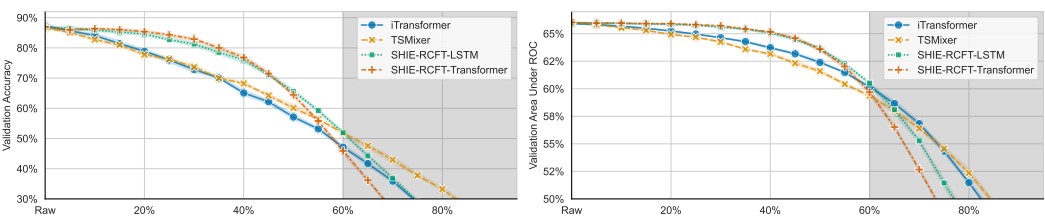

Figure 9: Physical Activity Monitoring (ablation): missingness–metric curves. MCAR missingness from $0\%$ ("raw") to $95\%$. "Raw" refers to original, uncorrupted sequences. Shaded bands denote $95\%$ confidence intervals.

### F.3.3 ROOM OCCUPANCY ESTIMATION

This appendix section provides the complete results for *Room Occupancy Estimation*. Table 6 reports MR-AUC at $\alpha$ for *E-MAE* and *Accuracy*. Figure 10 shows the corresponding missingness–metric

curves (ablation), and Fig. 11 compares our SHIE–RCFT variants to the SOTA baselines *iTransformer* and *TSMixer* under the same MCAR sweep from $0\%$ ("raw") to $95\%$.

Table 6: Full MR-AUC results for the Room Occupancy Estimation dataset. Metrics reported are $E$-MAE (lower is better) and Accuracy (higher is better). $\alpha$ is the MR-AUC truncation parameter indicating missingness up to that level (see Sec. 5.1). "S–R" denotes the combined use of the SHIE and RCFT modules.

| $\alpha$ | Metric | LSTM | | | | Transformer | | | | TSMixer | iTrans. |
|---|---|---|---|---|---|---|---|---|---|---|---|
| | | Vanilla | SHIE | RCFT | S-R | Vanilla | SHIE | RCFT | S-R | | |
| 0.25 | $E$-MAE | 0.017 | 0.017 | 0.018 | 0.019 | 0.025 | 0.023 | **0.016** | **0.016** | 0.337 | 0.143 |
| | Accuracy | **0.988** | 0.986 | 0.985 | 0.984 | 0.972 | 0.980 | 0.986 | 0.986 | 0.901 | 0.910 |
| 0.50 | $E$-MAE | 0.065 | **0.015** | 0.029 | 0.027 | 0.030 | 0.033 | 0.023 | 0.023 | 0.868 | 0.159 |
| | Accuracy | 0.952 | **0.988** | 0.979 | 0.980 | 0.971 | 0.976 | 0.983 | 0.983 | 0.801 | 0.907 |
| 0.75 | $E$-MAE | 0.223 | **0.025** | 0.044 | 0.036 | 0.040 | 0.044 | 0.033 | 0.031 | 1.232 | 0.128 |
| | Accuracy | 0.799 | **0.982** | 0.972 | 0.977 | 0.968 | 0.973 | 0.979 | 0.980 | 0.707 | 0.925 |

**Notation:** $E$ = expected occupant count; see App. C.3 for details.

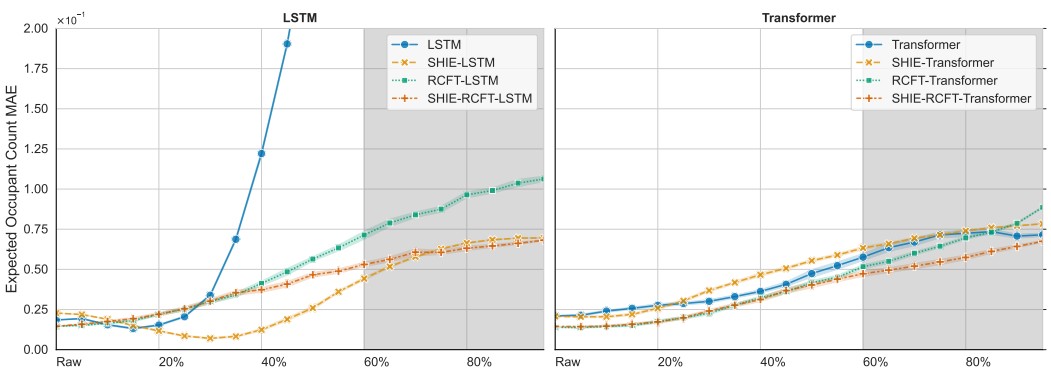

Figure 10: Room Occupancy Estimation (ablation): missingness–metric curves for Expected MAE (E-MAE; lower is better). MCAR missingness from $0\%$ ("raw") to $95\%$; "raw" denotes original, uncorrupted sequences. Shaded bands denote $95\%$ confidence intervals.

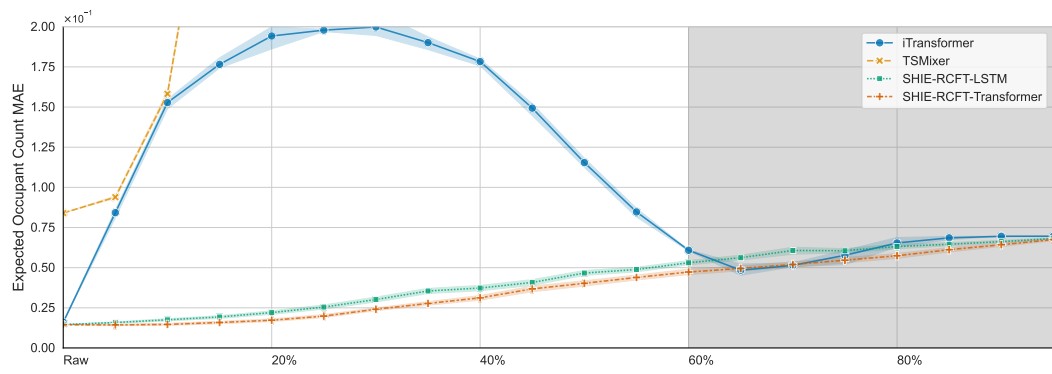

Figure 11: Room Occupancy Estimation (comparison): missingness–metric curves for Expected MAE (E-MAE; lower is better), comparing our method against iTransformer and TSMixer under the same protocol. MCAR missingness from $0\%$ ("raw") to $95\%$; "raw" denotes original, uncorrupted sequences. Shaded bands denote $95\%$ confidence intervals.

### F.3.4 COGNITIVE LOAD ESTIMATION

This appendix section provides the complete results for *Cognitive Load Estimation*. Table 7 reports MR-AUC at $\alpha$ for *Accuracy* and *AU-ROC*. Figure 12 shows the corresponding missingness–metric curves (ablation), and Fig. 13 compares our SHIE–RCFT variants to the SOTA baselines *CroSSL*, *MBT*, and *TSMixer* under the same MCAR sweep from $0\%$ ("raw") to $95\%$.

Table 7: Full MR-AUC results for the Cognitive Load Estimation dataset. Metrics reported are Accuracy and AU-ROC (higher is better). $\alpha$ is the MR-AUC truncation parameter indicating missingness up to that level (see Sec. 5.1). "S–R" denotes the combined use of the SHIE and RCFT modules.

| $\alpha$ | Metric | LSTM | | | | Transformer | | | | TSMixer | MBT | CroSSL |
| --- | --- | --- | --- | --- | --- | --- | --- | --- | --- | --- | --- | --- |
| | | Vanilla | SHIE | RCFT | S-R | Vanilla | SHIE | RCFT | S-R | | | |
| 0.25 | Accuracy | 0.627 | 0.650 | 0.659 | 0.645 | 0.570 | 0.566 | 0.659 | **0.712** | 0.679 | 0.663 | 0.584 |
| | AU-ROC | 0.702 | 0.731 | 0.723 | 0.718 | 0.601 | 0.620 | 0.715 | **0.806** | 0.762 | 0.697 | 0.622 |
| 0.50 | Accuracy | 0.612 | 0.641 | 0.660 | 0.628 | 0.549 | 0.557 | 0.624 | **0.680** | 0.668 | 0.614 | 0.563 |
| | AU-ROC | 0.675 | 0.711 | 0.720 | 0.716 | 0.576 | 0.620 | 0.685 | **0.792** | 0.754 | 0.684 | 0.606 |
| 0.75 | Accuracy | 0.586 | 0.618 | 0.644 | 0.604 | 0.538 | 0.549 | 0.590 | **0.653** | 0.648 | 0.562 | 0.552 |
| | AU-ROC | 0.654 | 0.692 | 0.710 | 0.705 | 0.565 | 0.623 | 0.648 | **0.770** | 0.746 | 0.664 | 0.594 |

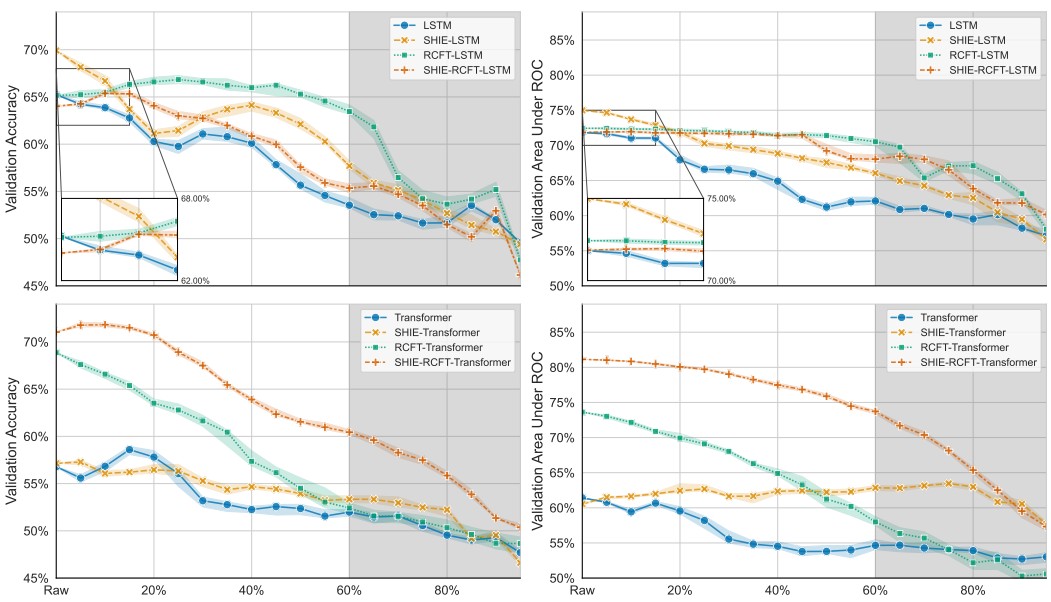

Figure 12: Cognitive Load (ablation): missingness–metric curves for Accuracy and AU-ROC (higher is better). MCAR missingness from $0\%$ ("raw") to $95\%$; "raw" denotes original, uncorrupted sequences. Shaded bands denote $95\%$ confidence intervals.

## G ADDITION STATEMENT AND INFORMATION

### G.1 IMPACT STATEMENT

This work presents a domain-agnostic framework for multi-sensor time series prediction under missing data. Because the technique is application-independent, its societal effects, positive or negative, will be driven by the downstream tasks to which practitioners deploy it. In beneficial settings (e.g., smarter energy grids, improved building efficiency, or real-time health monitoring), the method could reduce waste, enhance safety, and support timely decision making. In contrast, if integrated into high-stakes surveillance or automation without adequate oversight, it can amplify

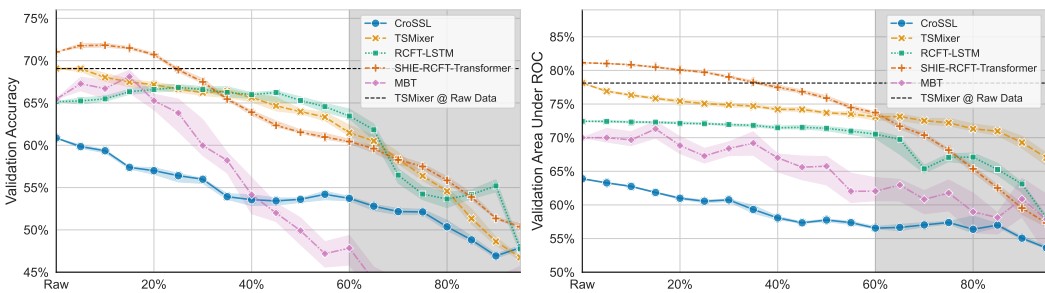

Figure 13: Cognitive Load (comparison): missingness–metric curves comparing our method to iTransformer and TSMixer. MCAR missingness from $0\%$ ("raw") to $95\%$; "raw" denotes original, uncorrupted sequences. Horizontal dashed lines indicate TSMixer performance on uncorrupted data. Shaded bands denote $95\%$ confidence intervals.

privacy concerns or propagate errors that disproportionately affect certain groups. The model itself does not include domain-specific assumptions, but we emphasize the importance of application-level safeguards, such as data governance policies, fairness testing, and human-in-the-loop monitoring, to ensure responsible use.

## G.2 EXAMPLE TRAINING CONFIGURATION

We release full training scripts and per-experiment configuration files in the supplementary material; each task–dataset pair has its own hyper-parameter grid. To illustrate the format, Table 8 lists one concrete instance—the SHIE-RCFT-Transformer used for the 4583-node voltage-estimation task—while all other settings can be found in the supplement materials.

| Parameter | Value |
| --- | --- |
| Attention dropout | 0.1 |
| Attention embedding dimension | 256 |
| Feed-forward layer size | 2048 |
| Model Dimension | 1024 |
| Feed-forward dropout | 0.1 |
| Attention heads | 2 |
| Query/key dimension | 256 |
| Query ratio | 2 |
| Activation | GELU |
| RCFT layers | 6 |
| Output channel dimension | 16 |
| Output shape | $T \times 8515 \times 2$ |
| SHIE sub-channel dimension | 16 |

Table 8: Example hyper-parameters for the **4583-node** grid-voltage experiment.

## G.3 COMPUTATION RESOURCES

All results are generated on a DGX A100 system with 8 GPUs. The total system memory is 1.5 TB, and the total of 640 GB GPU memory. The training execution time for most models and datasets is within 1 hour. Grid Estimation with 4853-node requires about 3 hours per model per run. Cognitive Load Estimation takes around 10 minutes per model per run. Room occupancy takes 1 hour to run. Physical Activity Monitoring takes 2 hours to run. RCFT scalability analysis required substantially longer time of about 7 hours per model per run. iTransformer and TSMixer take substantial longer times to run on PAMP2, compared to SHIE-RCFT, with up to 3 times longer runtime than the base model.

# H   USAGE OF LARGE LANGUAGE MODEL

In preparing this manuscript, we made limited, non-substantive use of a large language model (LLM) for editorial polishing and mechanical formatting (e.g., rephrasing for clarity and transcribing results from CSV files into LaTeX tables). We also used the LLM as a secondary safeguard to check anonymity compliance (e.g., scanning for accidental identifiers) and to flag minor code issues after human review. All algorithms, datasets, experimental designs, analyses, and conclusions were conceived and executed by the authors, and all code and table entries were verified against the underlying results.

