# OpenReview forum: "Causal Representation Learning on Degraded Multi‑Sensor Streams"
_ICLR.cc/2026/Conference — Submitted to ICLR 2026_

### Official Review · Reviewer_JMYZ · 2025-10-27

**Soundness:** 3
**Presentation:** 3
**Contribution:** 3
**Rating:** 4
**Confidence:** 2

**Summary:**

The paper introduces a causal multimodal fusion framework for time-series inference under missing or corrupted sensor data. It proposes two key modules. 1- SHIE, which structures each sensor’s subchannels to localize the effect of missingness, and 2- RCFT, which alternates causal temporal modeling and cross-modal attention to iteratively refine representations across modalities. These components form a end to end model capable of real time, streaming inference while keeping causal constraints. The approach is tested on diverse datasets including physical activity, smart grid estimation, and cognitive monitoring showing improved robustness and accuracy compared to prior multimodal baselines.

**Strengths:**

- The paper introduces two simple and effective plug and play components that are easily integrated into a wide range of existing causal or sequential models. This modular design enhances flexibility and show the potential for applying different BBs.

- The paper is easy to follow with clearly making notation, and motivation of each component. The logical flow from problem formulation to architectural design makes the ideas easier to access and justify.

- The paper provides the code with an easy-to-follow instruction that makes the following up process much easier. I liked that.
This level of transparency is valuable for the research community.

- They provide relatively complete ablation study with covering different ratios of data corruption and missingness. This thorough analysis helps validate the robustness and general effectiveness of their model.

-  The appendix is useful and provides additional structural details along with the additional results, data format, and noise insertion setting. These supplementary materials add clarity and reproducibility to the overall work which is valuable.

**Weaknesses:**

- The causal modeling is motivated for real-time or streaming applications, and comparing under this constraint is reasonable.
But in some settings we may have access to the full observation sequence (e.g., offline analysis or retrospective inference). In that case, how to use future context to refine your latents? kinda smoothing or bidirectional inference variant would be feasible and likely beneficial in this case.
I suggest either including a full-sequence inference experiment or, at minimum, clarifying how the current causal architecture could be extended to support non-causal or bidirectional inference. This can make your work applicable to broader apps.

- The paper (in their intro) argues that two-stage smoothing or imputation pipelines can not be trained end-to-end and propagate errors to the predictor. This is true for classical modular pipelines, but recent differentiable smoothing frameworks (e.g., variational approaches, diffusion-based imputers, or RNN-based bidirectional models) allow joint, end-to-end optimization. Please clarifying this in draft.

- The model assigns separate SHIE and RCFT modules to each modality, which preserves modality-specific representations but can make scalability issues. The computational and parameter complexity grow with the number of modalities, making the approach potentially inefficient for systems with many sensors or high modality diversity. For this concern, the paper does not include an explicit complexity analysis.

- The reported results lack error bounds or standard deviations. Although in appendix (line 1232) running trials with different random seeds are stated, only single-point error and accuracy values are presented in the tables. Without reporting variability across runs, it is difficult to assess the robustness of the method to initialization or its statistical reliability.

- Although I found detailed hyperparameter values in the released code, the paper does not clearly describe how these values were determined (e.g., via grid search, random search, or heuristic tuning). Providing this information would improve reproducibility and ensure a more balanced comparison across methods.

- Both UniTime [1] and PowerPM [2] share conceptual overlap with this work and can strengthen the benchmark evaluation (currently the benchamrks are four: TSMixer, iTrans, CroSSL and MBT).
UniTime addresses cross-domain time-series modeling with heterogeneous inputs, similar to the this paper goal of multimodal inference across varying feature dimensions. PowerPM targets large-scale electricity time-series with hierarchical and temporal dependencies, again similar with this paper structure, causal modeling of smart-grid data. Including these two models as additional benchmarks (or at least arguing how they are related to this work and how they are expected to perform in your exps) would provide a more diverse and representative comparison.

- All modalities used in the experiments seems to continuous valued signals, such as physical or physiological sensor readings. It remains unclear how the proposed framework would handle categorical or discrete modalities. If such an extension is possible, it would be useful to explain how categorical inputs are embedded and fused. Otherwise, this limitation should be explicitly acknowledged in the paper.

[1] - Liu, Xu, et al. "Unitime: A language-empowered unified model for cross-domain time series forecasting." Proceedings of the ACM Web Conference 2024. 2024.

[2] - Tu, Shihao, et al. "Powerpm: Foundation model for power systems." Advances in Neural Information Processing Systems 37 (2024): 115233-115260.

**Questions:**

Thank you for your work and all the efforts. I have some questions about the paper:

- How can the proposed model be extended to full (non-causal) inference, where future observations are also used to refine latents or predictions? If such an extension is feasible, please consider including an experiment. If no, clearly state in the draft what structural and objective modifications would be required to achieve this, and mention a reasonable training objective or procedure for that setting.

- Originally you make notation that modalities are shown with $n$ (line 113). Later you switch to $m$ to note modality (e.g. line 697). Could you make the notation consistent? Although the modality branch is representing each modality that finally is gonna build the output, but you named it 'task': $m \in {1,...,M}$ before.

- In line 714, why there is _m index for MHA? MHA is a single softmax operator over your Q,V,K without weight parameters, shared for all of your queries, right? Then, this notation can cause confusion that you have M separate MHA blocks rather than a shared one.

- Your model assigns independent SHIE and RCFT modules to each modality, which preserves modality-specific semantics but also increases parameter count and computational cost linearly (or worse) with the number of modalities. increase of accuracy is intuitive in this case, but it raises questions about scalability. Could you include a complexity analysis, either empirical (e.g., runtime scaling with modality count) or theoretical (e.g., asymptotic time and memory complexity in $\mathcal{O}(.)$ format )? it clarifies how the model’s cost grows with modality size.

- In your results, the S-R (T) variant substantially outperforms the TSMixer baseline in terms of error (e.g., your error of 1.191 is up to eight times lower than TSMixer in the Physical Activity Monitoring task). However, the corresponding accuracy improvement seems marginal (onnly a few percentage points). Could you provide an explanation for this?

- Why there is not standard deviation or error bound reported in tables? Could you add it with $\pm$ sign?

- Could you add a detailed explanation of the hyper parameter optimization? How did you come up with the final set of HPs? Same for the baselines, did you use the original HPs listed in their original drafts? Or did you follow a HPs optimization recipe for the baselines as well?

- Could you also explain how your structure can be extended to both categorical-continuous modalities inputs? It helps the reader to understand the scope of your work better.

---

> ### Author Response · Authors · 2025-11-19
> **Response Part 1**
>
> **Reviewer asks about the potential of our method extending to non-causal situations.**
> We thank the reviewer for the comment. When full sequences are available, the architecture can be extend by (1) removing the causal mask or (2) using a bidirectional inference module to incorporate future information.
> Meanwhile, we would like to point out that we tested on one non-causal problem (Cognitive Load Estimation) with a casual constraint (partially addressing this comment in terms of non-causal evaluation), our method still works reasonably well compare other traditional and SOTA baselines.
>
> **Reviewer raises concern about our description of imputation-based methods.**
>
> We thank the reviewer for this insightful comment. You are correct. Our statement in the introduction was intended to refer to *classical* two-stage pipelines where a fixed, non-differentiable smoother/imputer is run before prediction. We agree that recent approaches (e.g., BRITS, latent VAE/ODE models, diffusion-based imputers, bidirectional RNNs) enable differentiable, end-to-end training with downstream predictors, and we will adjust the wording and add citations to make this clear.
>
> Our focus in this paper, however, is on strictly causal, real-time estimation. Many of the strongest end-to-end imputers exploit bidirectional or full-sequence context, which makes them less directly applicable in streaming settings where the most recent $N$ points may be missing and the model must effectively extrapolate rather than interpolate. Designing a unified imputation/extrapolation--plus--prediction module that is both end-to-end differentiable and strictly causal is certainly possible, but substantially more complex, and lies outside the scope of the present work.
>
> Finally, on one of our non-causal tasks (Cognitive Load), the SHIE-RCFT-Transformer with 20\% MCAR *still* outperforms SOTA baselines evaluated on clean data (0\% MCAR), as visible from our robustness scans. This suggests that even with accurate imputations, robustness of the learned representation itself is crucial, and that simply "perfectly" imputing a clean signal for a traditional model may not be sufficient. We will revise our draft to clarify that our critique is aimed at classical modular pipelines and that our approach is complementary to modern differentiable imputers.
>
> **Reviewer raises concern about lack of error bounds and standard deviations.**
> We thank the reviewer for the comment. While this is missing from Table-2, we *did use error bars* for full results in the Appendix-F.3. The main reason we did not include deviation or range values directly in Table-2 is that the table is already very tight. Adding additional columns for variability would require us to remove significant digits from the mean errors, which would make the comparisons between models less precise and less readable. At the same time, the run-to-run variation of each model is much smaller than the differences between models: with the number of digits we can reasonably show in the main text, the deviation entries would mostly appear as zeros (or values with the same order of magnitude as the mean error), and thus provide limited additional insight.
>
> Appendix~F.3 reports the full MCAR scan. In all of these figures, the shaded bands denote the 95\% confidence intervals across runs.  Most of these bands are so narrow that they are almost invisible at the default scale and only become noticeable when zooming in, which indicates that the evaluation noise across runs is negligible compared to the differences between models.
>
> For these reasons, we believe that adding deviation columns to the main table would mainly harm readability without changing the conclusions. However, we are happy to provide the full MCAR-scan run logs (per-run results for all models) as supplementary material, so that readers can compute deviations or generate their own figures if desired. Moreover, as stated in the Reproducibility Statement, we provide our code, dataset, and hyperparameters so that interested readers can rerun the experiments and independently verify these statistics.
>
> **Reviewer raises concern about missing hyperparameter clarification.**
>
> We thank the reviewer for the comment. We did not conduct HP search. Parameters are tuned by heuristic. We set width approximate 3 times of input dimension with minimal 128. And using 6 layers for Power Grid Estimation and 4 for others. Since Power Grid Estimation has much larger input dimension.
>
> Also, scalability experiments indicate that performance can further improve through HP tuning/search. But here we focus more on comparison of traditional baseline and SOTA baseline under similar settings.

---

> > ### Author Response · Authors · 2025-11-19
> > **Response Part 2**
> >
> > **Reviewer pointed out UniTime and PowerPM might be potential related work**.
> >
> > We thank the reviewer for pointing these references out. We will add these two references into our related work.
> >
> > **Reviewer commented that we do not consider how to deal with discrete valued sensor data.**
> >
> > We thank the reviewer for this comment. The design of our subchannel input is meant to organize inputs individually (but grouped by sensor type), thus learning a "knowledge of sensors" for a given sensor group. Thus, discrete or categorical inputs can theoretically naturally be organized with one-hot-encoding under the same subchannel (learning a continuous embedding from the OHE and other grouped sensors). We have added this additional description in the appendix of the paper.
> >
> > **Reviewer points out several typos and inconsistent conventions**
> >
> > We thank the reviewer for this detailed comment. We have fixed all inconsistencies.
> >
> > **Reviewer asked about a specific error value and inconsistent result in the table from dataset to dataset.**
> >
> > We thank the reviewer for carefully reading our results.
> > The key reason for the apparent discrepancy is that the large error reduction the reviewer cites (e.g., error of $1.191$ being up to eight times lower than TSMixer) appeared in experiments with **Power Grid Estimation** dataset, which is a regression problem; while the corresponding "only a few percentage points" improvement is measured on **PAMAP2**, a classification task where we report Accuracy.
> > On PAMAP2, missing data typically changes a single prediction from right to wrong (or vice versa). Each example contributes equally to Accuracy, so even if S-R (T) corrects many such cases, the overall Accuracy can increase only by a few percentage points.
> > Power Grid Estimation behaves very differently. In this dataset the grid has two operating topologies (Section 4.1 line 293) : several buses are switched on or off by breakers, but this topology change is not provided as input. The model has to infer it from the power-flow relationships. When an entry is marked as missing, the model must determine whether this bus is on or off. When a bus is switched off, its true voltage is essentially $0$ (per unit), and when it is switched on, it is close to $1.0$. When the estimation stays on the correct side, the typical per-bus error is around $10^{-3}$. However, if the model misidentifies the topology and treats a switched-off bus as if it were on (or vice versa), the error at that bus jumps to the order of $1.0$, i.e., roughly three orders of magnitude larger.
> > When models misjudged the topology state, it creates a very large local error. Causing the aggregated regression error (e.g., MAE/RMSE) to increase significantly. Our S-R (T) variant reduces exactly these topology-related failures, thus, the overall regression error can drop by a large factor, even though the fraction of affected time steps is relatively small.
> > We illustrate this effect in Appendix~F.2, where we plot per-bus residuals. Most of the large mistakes made at the start of the study are concentrated in the busses $75$ and $125$. When the baseline misinterprets the operating mode at these buses, it results in very large errors, while S-R (T) substantially reduces these outliers.
> > This leverage effect is the main reason why the improvement on PAMAP2 appears marginal in terms of Accuracy, whereas the improvement on Power Grid Estimation appears very large in terms of regression error.
> >
> > **Reviewer also asked about computation efficiency analysis**
> >
> > We have a detailed response to this question in the section for reviewer yjjw.

---

### Official Review · Reviewer_CjVi · 2025-10-28

**Soundness:** 3
**Presentation:** 3
**Contribution:** 3
**Rating:** 6
**Confidence:** 3

**Summary:**

This paper targets causal, real-time fusion of multi-sensor streams where inputs can be missing or degraded. The authors propose two plug-in modules that can be attached to unidirectional backbones (LSTM or causal Transformer): (i) Subchannel Hierarchical Input Embedding (SHIE): embeds each sub-sensor (e.g., accelerometer axes) independently and then aggregates at the channel level, so that missing/corrupted values perturb only a local slice of the representation. (ii) Repetitive Cross-Modal Fusion Transformer (RCFT): maintains one branch per modality and alternates a causal temporal update with modality-wise attention at every layer to iteratively exchange information across streams while respecting causality. They evaluate on four tasks—grid voltage estimation (84- and 4583-node systems), physical activity monitoring (PAMAP2, wrist-only), room occupancy estimation, and cognitive load classification—injecting degradations under an MCAR policy and sweeping missingness up to 95%.

**Strengths:**

+ The paper is explicitly about causal, streaming inference with missing and degraded inputs, which is an important and practical setting often sidestepped by imputation pipelines that use future context.

+ SHIE and RCFT are architecture-agnostic modules for LSTMs or causal Transformers; the design keeps per-modality identity, enabling iterative fusion without early entanglement.

+ SHIE’s locality preservation and RCFT’s message-passing style, with causal masking and per-layer cross-modal attention, are motivated and contrasted with early/late fusion and window forecasters.

**Weaknesses:**

- Training and eval use MCAR, with temporally clustered bursts, but many deployments exhibit MAR/MNAR patterns tied to environment or device dynamics. Results may overstate robustness when missingness is not feature- or context-dependent;

- Although tasks span domains, they are primarily tabular sensor streams. Vision/audio or high-bandwidth modalities are not tested, limiting claims of domain-agnosticism in rich-media settings;

**Questions:**

1. Have you tested RCFT/SHIE under MAR (feature- or state-dependent missingness) or MNAR? If not, can you share preliminary results or hypothesize failure modes vs. MCAR?

2. Beyond architecture locality, have you tried auxiliary objectives (e.g., self-supervised outlier scoring per subchannel) to explicitly teach the model to down-weight corrupted readings during RCFT fusion?

3. How sensitive is performance to how often cross-modal attention is performed (e.g., every L layers vs. every layer)? Any diminishing returns from fusion density at fixed depth?

---

> ### Author Response · Authors · 2025-11-19
> **Response Part 1**
>
> **Reviewer raises concern about vision/audio or high-bandwidth modalities not being tested, limiting claims of domain-agnosticism in rich-media settings**
>
> We thank the reviewer for the comment.
> We agree that our experiments focus on multivariate sensor streams and do not directly include semantically rich modalities such as raw video or audio, which typically are processed in more "domain specific" ways. Our goal in this paper is to study robust representation learning under a causal constraint for degraded classical sensor streams, so we concentrated on this setting.
> We note, however, that one of the systems we consider is already very high-dimensional: in the 4,583-node Power Grid estimation task, each time step has on the order of 16,000 input/output features, which is comparable in size to a VGA frame ($640 \times 480$), and the model is trained with only about 4,000 available time points. This illustrates that our approach can handle large-scale input spaces.
>
> **Reviewer commented that that beyond architecture locality, auxiliary objectives (e.g., self-supervised outlier scoring per subchannel) that explicitly training the model to down-weight corrupted readings during RCFT fusion could be beneficial.**
>
> We thank the reviewer for the comment--this is an excellent idea for follow up work.
> We have not explored auxiliary objectives such as self-supervised outlier scoring per subchannel in the current work. In principle, however, one could introduce paired noisy and noise-free inputs and use a self-supervised objective (e.g., via coupling layers or earlier stages of SHIE--RCFT) to pull their representations closer together, so that the network learns a more noise-invariant representation and can better capture the relationship between clean and corrupted patterns.
>
> A similar idea was carried out by Barnett, Naim, et al., who recently published a Joint Embedding Predictive Architecture (JEPA) inspired Transformer Architecture for cognitive load estimation (similar task, but different dataset), which made use of self-supervised learning. The reported accuracy is 70.59$\%$ with an AUC of 0.696, which are worse than our best models. [1]
>
> In this paper, we focus on the core dynamics of the SHIE--RCFT architecture and its behavior under different degradation conditions, without additional objectives that might confound the analysis. We view such paired self-supervised losses as a promising add-on and provides a very interesting follow up work.
>
> [1] Barnett, Naim, et al. "Generalizing Classification of Pilot Workload: Transfer Learning versus a JEPA-Inspired Transformer Architecture." International Journal of Aviation, Aeronautics, and Aerospace 12.1 (2025): 2.
>
> **Reviewer asks about sensitivity of performance to how often cross-modal attention is performed (e.g., every L layers vs. every layer) and potential diminishing returns from fusion density at fixed depth.**
>
> We thank the reviewer for the comment and appreciate the suggestion.
> We have not performed a systematic study of how performance varies with the *frequency* of cross-modal attention. In the current SHIE--RCFT design, fusion is applied at every layer by alternating temporal attention and cross-modal attention, and most of our main experiments use relatively shallow models (4 layers). For larger models, this would be a natural way to reduce computation while retaining most of the benefit of fusion, and we see this as an interesting direction for future work.
>
> We did, however, run depth-scaling experiments (Appendix~F.1) on RCFT-based architectures. On the 84-node Power Grid estimation task, a Transformer backbone with RCFT was scaled up to 48 layers, with cross-modal fusion on each layer. In that setting, performance continued to improve as depth increased; the 48-layer model (about 1.7 billion parameters, roughly 50$\times$ larger training dataset) achieved the best results we observed. While the marginal benefit per additional layer decreases, we did not observe a clear performance drop or early saturation that would indicate strong diminishing returns within the depth range we tested.
>
> **Reviewer also raised concern about MCAR versus MAR/MNAR noise**
>
> We address is this in the reviewer yjjw section.

---

### Official Review · Reviewer_pN1V · 2025-11-01

**Soundness:** 1
**Presentation:** 1
**Contribution:** 2
**Rating:** 2
**Confidence:** 5

**Summary:**

This paper proposes two "plug-in" modules, SHIE and RCFT, intended to improve the robustness of causal sequence models (like LSTMs or Transformers) for multi-sensor time-series tasks. The stated goal is to handle degraded and missing data in real-time.

   1. SHIE (Subchannel Hierarchical Input Embedding) processes sensor inputs by first embedding fine-grained "subchannels" (e.g., X/Y/Z axes of an accelerometer) before aggregating them, with the goal of isolating local sensor noise.

   2. RCFT (Repetitive Cross-Modal Fusion Transformer) maintains a separate branch of computation for each sensor modality and performs iterative, layer-by-layer fusion using cross-attention at each time step.

The authors evaluate these modules by adding them to LSTM and Causal Transformer backbones and testing them on four multi-sensor datasets. The core of the evaluation is a robustness test where models trained with 5% missing data are evaluated on test sets with up to 95% missing data. The paper claims this approach significantly outperforms both the vanilla backbones and recent SOTA models like iTransformer, TSMixer, and CroSSL.

**Strengths:**

1. The ideas behind isolating local noise (SHIE) and iterative refinement (RCFT) over one-shot fusion are correct and well-motivated
2. The paper does successfully demonstrate that a vanilla Causal Transformer or LSTM is insufficient for this task and that the RCFT module provides a significant performance boost (Table 2).
3. MR-AUC Metric: The proposed MR-AUC metric is a sensible and useful way to summarize model robustness across a range of degradation levels

**Weaknesses:**

This paper suffers from two major, disqualifying flaws: (1) a misleading title and framing that misrepresents the paper's contribution, and (2) a fundamentally unfair experimental comparison that invalidates its primary claims of state-of-the-art performance.

1.  **Misleading Title and Unfair Experimental Framing:**
    * **The title is a misnomer.** The paper is titled "**Causal Representation Learning** on Degraded Multi-Sensor Streams." This is a significant misrepresentation. The field of "Causal Representation Learning" (CRL) is a specific, well-defined area of research focused on discovering latent representations that capture the underlying causal graph or structural causal model (SCM) of a system. This field, (e.g., Schölkopf, 2019; Arjovsky et al., 2019; Locatello et al, 2021), aims to learn features that are invariant to interventions or domain shifts, often for counterfactual reasoning. This paper does **not** engage with this field in any way. It builds a *causal temporal model* one that respects the arrow of time (i.e., no lookahead). This is a standard constraint for any real-time system and is **not** "Causal Representation Learning." This misleading title seems intended to borrow prestige from a popular but unrelated field.
    * **This misrepresentation extends to the SOTA comparison.** The paper's headline claim of outperforming iTransformer, TSMixer, and MBT is invalid. The authors' method is a **causal, streaming, per-step** (sequence-to-one) model. The baselines it compares against are **window-to-window**, non-recurrent forecasters. As the authors admit (Appendix A.3), these models are not designed for this task. "Adapting" them by appending a linear layer (Section 5.2) hobbles them and guarantees their failure, especially as they have no recurrent state to fall back on when inputs are missing. The "collapse" of SOTA models (e.g., Fig. 7) does not demonstrate the superiority of SHIE/RCFT; it demonstrates that the wrong tool was used for the job. This is a classic "apples-to-oranges" comparison and is not valid.

2.  **Incremental Novelty and Unclear Contributions:**
    * **SHIE:** The idea of embedding sub-components (subchannels) before aggregation is a standard hierarchical processing technique. It is a sensible engineering choice (akin to patching) but is not a novel contribution.
    * **RCFT:** Iterative, layer-by-layer fusion using cross-attention is a well-known pattern in multi-modal literature.
    * **The Ablation (Table 2) is Damning:** The paper's *only* valid comparison is the internal ablation. This shows that RCFT does almost all the work. For the 84-node Transformer (A-MAE*), the vanilla model scores 2.345. Adding SHIE-only brings this to 0.405 (a good improvement). But adding RCFT-only brings it to 0.224. The final combined model (S-R) scores 0.209. This demonstrates that RCFT is the primary driver, and the contribution of SHIE is marginal at best.

**Questions:**

1. Why is the paper titled *"Causal Representation Learning"* when it does not appear to engage with the main CRL literature (for example, learning invariant structural causal models) and instead focuses on a standard causal temporal model?

2. How can the reported SOTA comparisons (for example, against iTransformer) be considered valid when the baseline models are non-causal, window-based architectures that seem to have been adapted in ways that inherently disadvantage them in a streaming setting with high missingness?

3. Why is the most critical ablation missing, namely, the performance of a simple backbone combined with RCFT only? From Table 2, this component appears to account for nearly all the reported gains, which would make SHIE a minor contribution.

4. Following the previous question, why were more appropriate baselines — such as other robust, streaming, stateful models (for example, state-space models or recurrent architectures), not included for comparison?

---

> ### Author Response · Authors · 2025-11-19
> **Response Part 1**
>
> **Reviewer accuses authors of misleading title in attempt to borrow prestige from other field.**
> We acknowledge that "Causal Representation Learning (CRL)" is an established term. However, we had no intention to borrow prestige in an established field and are happy to adjust the title to make our intent explicit, for example to "Causally Constrained Representation Learning on Degraded Multi-Sensor Streams." We further point out that the term "causal" can be ambiguous in many contexts, between a temporal "causal constraint" on the estimator and "causal representation" as used in the CRL literature.
>
> In our paper, we use the term "causal" in the standard systems and signal-processing sense: at time $t$, the estimator may depend only on observations up to $t$, cannot use future observations, and past outputs are not revised once produced. Our work therefore focuses on "learning representations" under a temporal causality constraint, rather than learning a "causal representation" in the sense of inferring latent causal variables or counterfactual structure.
>
> **Reviewer argues that our comparisons are invalid.**
> We respectfully disagree with the claim that our comparison to iTransformer, TSMixer, and MBT is a misrepresentation or an "apples-to-oranges" evaluation. We include these models precisely because they are among the strongest SOTA architectures available for time-series modeling, and omitting them would weaken the empirical assessment of our contribution. Importantly, even after adapting them to the streaming causal setting required by our task (details below), they still outperform traditional causal baselines in our experiments by a substantial margin, so they remain strong and meaningful reference models rather than “hobbled” ones.
>
> Our proposed method itself is a Many-to-Many (NOT Many-to-One) architecture under a strict causal constraint. Existing SOTA methods are typically designed for forecasting or sequence classification, where full access to the input window is allowed and a Many-to-Many causal structure is not required. As discussed in Section-1 and formalized in Section-2, a real-time estimation system must respect the following constraint: we assume a $T$-lag input $X_{0:T} = (X_0, \dots, X_T)$ and outputs $Y_{0:T} = (Y_0, \dots, Y_T)$, where each $Y_i$ for $i = 0, \dots, T$ cannot access future inputs $X_j$ with $j > i$ and can only depend on past and current inputs, i.e., $Y_i = f_i(X_{0:i})$. In traditional LSTMs this causality is enforced naturally, and in standard Transformers it can be enforced via a causal mask, so they can be run in a Many-to-Many causal setting. In contrast, architectures such as iTransformer and TSMixer employ fully connected layers over the temporal dimension, which inherently allow access to future information in the window.
>
> Under these circumstances, if we naively used iTransformer or TSMixer in a Many-to-Many fashion, intermediate outputs would either be ill-defined under the causal constraint or would implicitly depend on future inputs, violating the problem setting. To adapt these models to the causal task without altering their core design, we apply a minimal and standard modification: we only add a linear layer at the end to collapse the sequence output to a single $Y_T$. Concretely, we use the iTransformer/TSMixer backbone to map the input window $X_{0:T} = (x_0, \dots, x_T)$ to hidden states $Z_{0:T} = (z_0, \dots, z_T)$, and then apply a linear layer over the time axis to $Z_{0:T}$ to obtain $y_T$. This turns the architecture into a Many-to-One model while ensuring that $y_T$ only accesses information from $X_{0:T}$, thus preserving causality and keeping the backbone as intact as possible given the task constraints.
>
> Conceptually, the main downside of this adaptation over a native causal model is computational cost: these non-causal models must recompute over the entire window when new data arrive, whereas fully causal Many-to-Many models can update incrementally. Our empirical results in Appendix~F focus on predictive performance and show that, when the evaluation noise matches the training noise, the adapted iTransformer and TSMixer still outperform traditional causal baselines by a large margin and behave much more similarly to our proposed model. The key empirical result is that our model maintains substantially better performance than iTransformer/TSMixer as noise levels increase beyond the training regime, indicating that our architecture is more robust to distribution shift and missing or corrupted inputs, which is precisely the main concern this work aims to address.

---

> ### Author Response · Authors · 2025-11-19
> **Response Part 2**
>
> **Reviewer argues that our baseline choices are inappropriate.**
> We thank the reviewer for the comment.
> The simple (vanilla) LSTM and Transformer backbones above are precisely the streaming, stateful baselines the reviewer refers to. Their results, with and without SHIE/RCFT, are reported in Table-2 and Appendix-F alongside the adapted iTransformer/TSMixer models. Across all datasets, the plain recurrent and Transformer baselines perform well below the adapted SOTA models, and adding SHIE/RCFT substantially improves their performance beyond SOTA, so the comparison includes both simple streaming architectures and strong non-causal SOTA models under the causal adaption.
>
> **Reviewer is concerned with limited novelty, inconsistent empirical gains and marginal improvements**.
> These are answered in responses to reviewer yjjw (novelty and Table-2 discussion) and in reviewer JMYZ (error value and inconsistent result discussion).

---

### Official Review · Reviewer_yjjw · 2025-11-01

**Soundness:** 3
**Presentation:** 3
**Contribution:** 2
**Rating:** 4
**Confidence:** 4

**Summary:**

The paper proposes two plug-in modules, Subchannel Hierarchical Input Embedding (SHIE) and Repetitive Cross-Modal Fusion Transformer (RCFT), to improve robustness in causal multi-sensor time-series modeling under missing or degraded inputs. SHIE builds pre-subchannel embeddings that localize noise effects and RCFT alternates causal temporal updates and modality-wise attention to enable iterative cross-sensor fusion.


Both modules can attach to any causal backbone (LSTM or causal Transformer). The authors evaluate on four datasets (power grid state estimation, PAMAP2 activity, room occupancy, and cognitive load) and compare performance against TSMixer, iTransformer, CroSSL, and MBT and introduce a new robustness metric MR-AUC (Missingness Robustness AUC).

**Strengths:**

S1. Clear motivation and framing. The problem of real-time degraded multi-sensor fusion is practical and under-explored.

S2. Methodologically sounds. The causal masking and modular plug-in design are coherent and reproducible.

S3. Broad evaluation. Four datasets, multiple backbones, and ablation studies are all included.

S4.Presentation quality. Writing, figures, and appendices are clear and detailed. Also included an anonymized OSF release for reproducibility.

**Weaknesses:**

exactly what is missing, etc.

W1.Limited novelty. RCFT is essentially a causal Transformer with modality-wise cross-attention, conceptually similar to iTransformer or standard multimodal Transformer. SHIE is a hierarchical embedding variant. Both are incremental architectural refinements rather than fundamentally new principles of causal modeling.

W2. Inconsistent empirical gains. Table 2 shows that SHIE and RCFT do not consistently improve performance.

- On PAMAP2, all variants perform identically (Accuracy ≈ 0.86, AU-ROC ≈ 0.66)

- On Cognitive Load, the plugins perform identically for LSTM.

- On Room Occupancy, the prediction accuracy improves slightly (0.3% for LSTM and 0.5% for Transformer)

- Only the grid estimation task shows strong numerical drops in MAE.

Therefore, the claim that both modules “consistently” enhance robustness is overstated. In addition, the results are reported as medians over 30 runs but without variance or significance testing. Small deltas could easily fall within random variation

W3. Weak justification of robustness setting. All experiments uses MCAR noise. Real-world degradation is typically MAR or MNAR correlated with context or sensor type. Thus, robustness to realistic failures remains unverified.

W4. No efficiency or latency evaluation. The paper emphasizes “lightweight plug-ins” and “streaming operation” but no FLOPs, parameter counts, or inference-time comparisons. It’s unclear whether the benefits justify the added architectural complexity.

**Questions:**

Q1: How sensitive is RCFT to the number of modalities and attention heads?

Q2: When do SHIE and RCFT fail to help (e.g., PAMAP2)?

Q3: What are the computational costs of adding both modules?

---

> ### Author Response · Authors · 2025-11-19
> **Response Part 1**
>
> We thank the reviewer for the detailed comments. We address each comment in sections, sometimes referring to responses given to other reviewers with similar critiques.
>
> **Reviewer raises concerns about limited novelty.**
> We agree that hierarchical processing is not, by itself, a novel idea. However, our contribution lies in analyzing how this specific hierarchical scheme (SHIE) behaves when applied to degraded multi-sensor streams, and in characterizing its dynamics when combined with the RCFT model. We argue that this is a simple idea, with extensive and thoughtful evaluation.
>
> To the best of our knowledge, RCFT's iterative layer-by-layer fusion and its explicit decoupling of temporal-wise and modality-wise processing is not a standard pattern in the multimodal literature. Most SOTA approaches, such as MBT and Perceiver, typically perform multimodal fusion by concatenating modalities along the temporal axis, and in our experiments their performance is suboptimal in the multi-sensor setting. If there are existing works that adopt a similar design to ours (i.e., decoupling attention over time and over modalities), we would be happy to include them in the related work section. A more detailed comparison is provided in Appendix A.3.
>
> **Reviewer raises concerns about inconsistent empirical gains and that improvements seem marginal.**
> The ablation in Table 2 reports only where training and evaluation noise levels are the same. This hides many of the advantages of SHIE-RCFT. A more comprehensive study across noise ratios is given in Appendix F3.1--F3.5. In short, RCFT is the main driver of increased performance and robustness, but SHIE plays a complementary role that is not captured by Table 2 alone: in the Power Grid experiments (Appendix F3.1), SHIE-RCFT and RCFT-only are similar when evaluation noise is very low (0--5%), yet SHIE-RCFT yields clear gains once noise exceeds the training level. Also in Physical Activity Monitoring (Appendix F3.2), RCFT-only lags behind some base models at low noise while SHIE-RCFT is still the best performer. Similarly, in Room Occupancy and Cognitive Load Estimation (Appendix F3.4--F3.5), SHIE-RCFT outperforms RCFT-only. Taken together, these results show that SHIE is not marginal: across datasets it systematically strengthens RCFT and, in the PAMAP2 case, prevents RCFT-only from degrading at low noise, so that the combined SHIE-RCFT architecture delivers robust performance across a range of noise conditions.

---

> ### Author Response · Authors · 2025-11-19
> **Response Part 2**
>
> **Reviewers raised concern about the study not experimenting with MAR/MNAR noise. Thus, robustness to realistic failures remains unverified.**
> We address this comment with two points:
> 1. True MAR/MNAR would require additional information about the datasets that is difficult or impossible for us to obtain.
> 2.  our noise is temporal clustered, such that it is representative of sensor data. Each argument is given in turn:
>
> We agree that real deployments can exhibit MAR/MNAR patterns. In our setting, however, a *realistic* MAR/MNAR mechanism would require parameterizing missingness using actual failure statistics from the underlying systems, since such mechanisms are typically value- and context-dependent. The public datasets we use do not provide this information. In the absence of such statistics, any MAR/MNAR scheme would be arbitrary and not meaningfully comparable across heterogeneous sensors and datasets; obtaining those statistics for each dataset would itself require a substantial, application-specific study that is beyond the scope of the present work.
>
> Our choice is therefore a controlled, application-agnostic stressor: **clustered MCAR on the inputs**, implemented via the sensor degradation protocol described in Section 3.4 and Appendix B. We use a data-agnostic MCAR mask that is temporally correlated, capturing the empirically common phenomenon that sensor noise and dropouts occur in bursts rather than as isolated points. In addition, we model unknown data corruption via value-dependent multiplicative noise with a similar clustered structure. This produces a structured degradation pattern on the inputs that is closer to typical sensor behavior than a randomly chosen MAR rule without supporting statistics.
>
> MAR/MNAR are problematic mainly because they introduce bias: missingness depends on the values, so the effective data distribution shifts. While we do not evaluate MAR/MNAR directly, we explicitly test robustness to distribution shifts in the missingness pattern. We train with a low-loss setting (5%) and then evaluate with the missing ratio from 0% (no synthetic loss) up to 95%. The strong performance over this range indicates reduced sensitivity to severe information loss and to shifts in the missingness statistics.
>
> We therefore do not claim general MAR/MNAR robustness in this paper. Instead, our sensor degradation simulation serves as a surrogate distribution shift test that probes MAR-like robustness issues. This combination both of clustered missingness and value-dependent corruption makes the test at least as challenging as many realistic MAR settings, and in some cases can be strictly harder in terms of information loss. In this sense, we respectfully disagree our robustness results are  weak: we test something closely related to robustness under realistic failures with the best information we have, and we view full MAR/MNAR evaluation as future, application-oriented work in settings where real missingness statistics can be obtained.

---

> ### Author Response · Authors · 2025-11-19
> **Response Part 3**
>
> **Reviewer raises concerns about missing computational efficiency analysis.**
> We did not conduct a systematic runtime study of our design, since our implementation is a training-focused library rather than an inference-optimized system. Instead, we can provide a theoretical complexity analysis. SHIE only adds three additional MLP projections over channels and subchannels, so its cost appears as a lower-order term in the overall $\mathcal{O}(\cdot)$ budget. RCFT introduces cross-time and cross-modality operations with complexity $\mathcal{O}(T^2 M + M^2 T)$, where $T$ is the time lag and $M$ is the number of modalities. In realistic setups, $M \ll T$, so one of the terms becomes negligible and the dominant cost is $\mathcal{O}(T^2 M)$; for a fixed application, $M$ is a constant, resulting in a total complexity bounded by $\mathcal{O}(T^2)$. If a real-world deployment uses a backbone with lower than quadratic temporal complexity (e.g., linear attention, LSTM, or Logformer), the overall complexity is less (such as $\mathcal{O}(M^2 T)$ for linear attention), further reducing the dependence on $T$ compared to a full quadratic-attention backbone.
>
> From partial empirical observations of training time, we find that both the base model and our most complicated model take roughly $2\text{–}3\times$ longer to train when using $2\text{–}3$ modalities. iTransformer can vary significantly with the input dimension size, since it applies attention over all channels rather than over grouped modalities, but its training time is usually about $0.8\sim 2\times$ that of our most complicated model.
>
> While all models are trained using rolling windows as a sampling strategy, during validation our causal predictors no longer require window recomputation, which amplifies the relative overhead of models that still depend on it. In the inference experiment where we vary the scan-missing ratio from $0$ to $0.95$, iTransformer and TSMixer typically take $3\sim 12\times$ longer than our most complicated model because they must recompute windows. In contrast, vanilla LSTM and Transformer usually require only $0.4\sim 1\times$ the runtime of our model. These estimates are not very precise, since when the runtime is short, dataset loading and model initialization time cannot be ignored. Nevertheless, the large margins suggest that recomputation is substantially more expensive than the additional complexity introduced by our model.
>
> We also note that there was concern about the term "lightweight" used in the paper. We agree with this and will remove this adjective from the paper because it is ambiguous. In the paper we use the term "lightweight" to describe the simplicity of the architecture, implementation, and its ease of integration with other models, rather than reduced computational runtime. In retrospect, this was a poor description.
>
> **Reviewer asked about sensitivity of RCFT to the number of modalities and attention heads.**
> For a given application the number of modalities is fixed, thus, it’s hard to test the sensitivity related to modalities while isolating other effect such as add or remove information. We did not experiment with different number of attention heads.
>
> **Reviewer asked situation where SHIE and RCFT fail to help (e.g., PAMAP2).**
> We thank the reviewer for the comment, but we would like to respectfully point out this question can potentially be interpret in a misleading way. While improvements seem marginal under low data loss, the gains become obvious as the missingness increases.

---

### Meta-Review · Area_Chair_4MmD · 2025-12-07

**Summary:**

This paper proposes two plug-in modules for causal, streaming multi-sensor time-series models under missing/degraded inputs: One embeds subchannels (e.g., accelerometer x/y/z) separately before aggregating to localise the effect of missing/noisy entries. Another maintains one branch per modality and alternates causal temporal updates with cross-modal attention to iteratively fuse sensor streams.

Reviewers generally agree that the problem setting is important, the design iseasy to plug into existing causal models, and the paper is overall clear and well-structured.

However, majority of reviewers express concern over limited novelty and comparison with existing methods. These concerns are not well addressed in rebuttal, therefore, the paper is recommended for rejection.

**Reviewer Concerns:**

One reviewer pN1V  strongly argues that the title “Causal Representation Learning on Degraded Multi-Sensor Streams” is misleading as there is a established research called causal representation learning. Authors suggested changing the title.

Concern shared among reviewers is limited novelty that proposed modules are incremental architectural refinements rather than fundamentally new principles. Rebuttal does not adequately address this concern.

Ablation results show RCFT accounts for most of the gains, with SHIE adding only marginal improvements. Rebuttal refers to Appendix showing the  SHIE plays a complementary role.

**Reviewer Scores:**

Reviewer yjjw not changed.

Reviewer pN1V not changed.


Reviewer CjVi  not changed.

Reviewer JMYZ might increase.

---

### Decision · Program_Chairs · 2026-01-26

Reject